# Ring walking as a regioselectivity control element in Pd-catalyzed C-N cross-coupling

Madeleine C. Deem[1], Joshua S. Derasp[1], Thomas C. Malig[1], Kea Legard[1], Curtis P. Berlinguette [1,2,3,4] & Jason E. Hein [1 ✉]

Ring walking is an important mechanistic phenomenon leveraged in many catalytic C-C bond forming reactions. However, ring walking has been scarcely studied under Buchwald-Hartwig amination conditions despite the importance of such transformations. An in-depth mechanistic study of the Buchwald-Hartwig amination is presented focussing on ligand effects on ring walking behavior. The ability of palladium catalysts to promote or inhibit ring walking is strongly influenced by the chelating nature of the ligand. In stark contrast, the resting state of the catalyst had no impact on ring walking behavior. Furthermore, the complexity of the targeted system enabled the differentiation between catalysts which undergo ring walking versus diffusion-controlled coupling. The insights gained in this study were leveraged to achieve desymmetrization of a tetrabrominated precursor. A small library of asymmetric 2,2′,7,7′-tetrakis[N,N-di(4-methoxyphenyl)amino]-9,9′spirobifluorene (Spir-oOMeTAD) derivatives were successfully synthesized using this strategy highlighting the ease with which libraries of these compounds can be accessed for screening.

[1] Department of Chemistry, University of British Columbia, Vancouver, BC V6T 1Z1, Canada. [2] Stewart Blusson Quantum Matter Institute, The University of British Columbia, 2355 East Mall, Vancouver, BC V6T 1Z4, Canada. [3] Department of Chemical & Biological Engineering, The University of British Columbia, 2360East Mall, Vancouver, BC V6T 1Z3, Canada. [4] Canadian Institute for Advanced Research (CIFAR), MaRS Innovation Centre, 661 University Ave. Suite 505, Toronto, ON M5G 1M1, Canada. ✉email: jhein@chem.ubc.ca

Triarylamines are characterized by low ionization potentials and an amorphous solid state, making them ideal candidates for organic electronic applications ranging from dye-sensitized solar cells to organic light-emitting diodes[1]. Optimization of both physical and electronic properties is achieved by rational design in combination with screening libraries of compounds. A characteristic example of this process is the development of the hole transport material SpiroOMeTAD (Fig. 1a). SpiroOMeTAD serves as the current gold standard hole transport material and has played a pivotal role in achieving power conversion efficiencies well above 20% in modern perovskite solar cells[2]. Significant research efforts have been deployed to improve the properties of SpiroOMeTAD by screening libraries of derivatives in which either the diarylamine or the spirocyclic core has been modified (Fig. 1b)[3–15]. Recent work from Chiykowski et al. demonstrated that asymmetric substitution of SpiroOMeTAD derivatives enabled independent optimization of physical and electronic properties (Fig. 1c)[16]. Thus, controlling substitution patterns in SpiroOMeTAD derivatives could significantly simplify rational optimization of material properties. However, accessing asymmetric species from polyhalogenated precursors using the Buchwald-Hartwig Amination (BHA) can be quite challenging[17,18].

Ring walking provides an opportunity to achieve difficult regioselectivity largely overlooked in small molecule synthesis. Ring walking occurs during cross-coupling when the catalyst remains bound to the pi system, migrates to a second electrophilic site, and undergoes another coupling event rather than dissociating from the pi system (Fig. 2, path A). The ability of catalysts to ring walk is an important mechanistic feature in chain transfer polymerizations (CTP) enabling access to polymers with controlled polydispersities[19–22]. Consequently, significant effort has been devoted to discovering nickel and, more recently, palladium catalysts that promote efficient CTPs for Kumada and Suzuki-type polymerizations[20]. Despite the importance of the BHA[23,24] and polytriarylamines[25], studies probing ring walking behavior under BHA conditions have remained virtually absent from the literature[26,27]. Furthermore, evidence for the mechanistic phenomenon of ring walking itself has remained largely circumstantial[19–22]. Mechanistic studies of CTPs have revealed turnover-limiting steps downstream of oxidative addition, precluding the ability to directly observe the short-lived catalyst pi complex[28–30].

It is also challenging to differentiate between authentic ring walking behavior and systems in which the catalyst does not remain bound to the pi system, yet manifest ring walking type behavior. In some systems intermediates have increased reactivity relative to the starting material, thus showing a preference for the formation of the product over the intermediate (Fig. 2, path B). These false positives are identified by obtaining time course data which necessarily shows the formation of an intermediate prior to product formation[31]. Alternatively, cases in which the catalyst dissociates from the pi system yet remains within the solvent sphere of the intermediate are impossible to differentiate from authentic ring walking (Fig. 2, path C)[20,32–35].

Herein we report a thorough mechanistic analysis of ligand effects on the synthesis of SpiroOMeTAD using the palladium catalyzed BHA. This study reveals the drastic effect ligand choice has on the evolution of intermediates throughout the reaction. These findings enable access to asymmetric SpiroOMeTAD

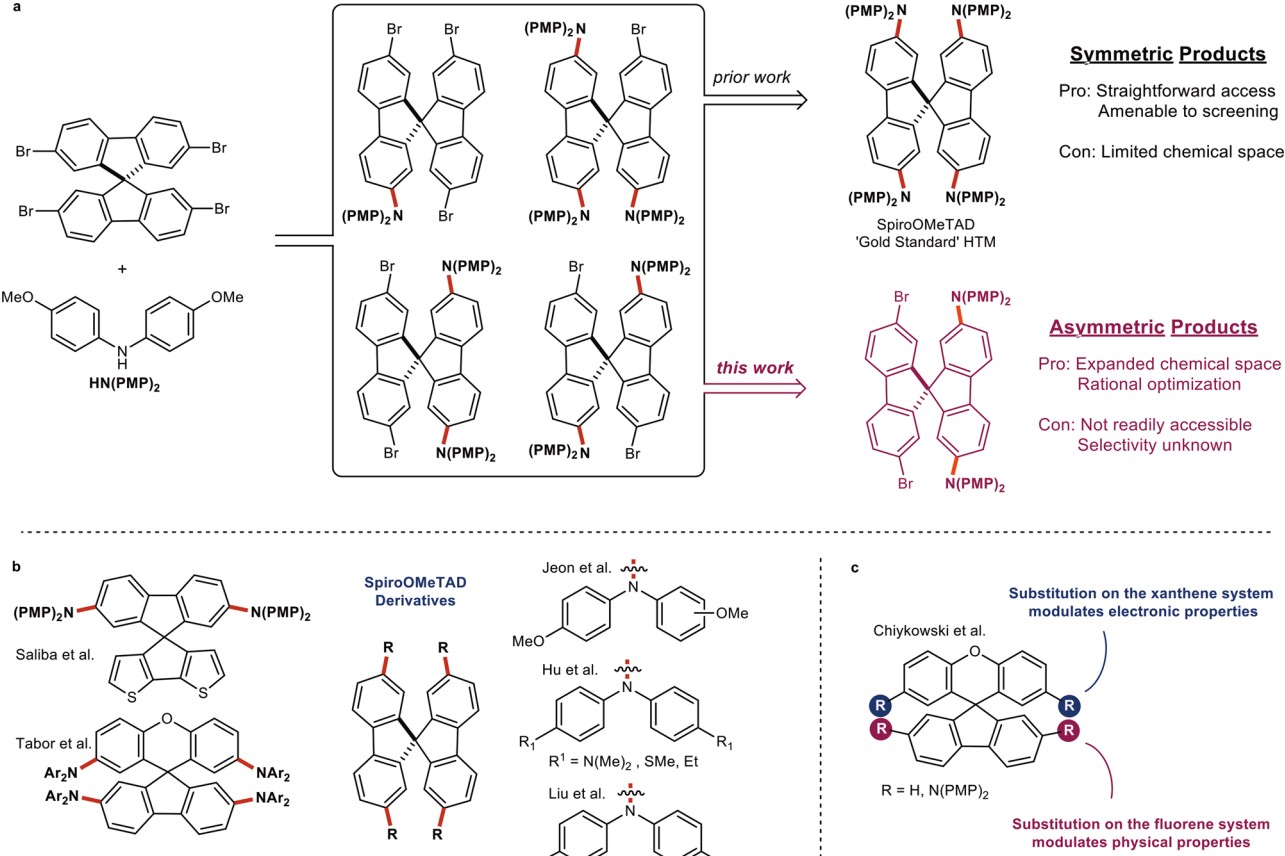

**Fig. 1 Importance of asymmetric hole transport derivatives. a** Standard synthetic procedure providing access to SpiroOMeTAD. **b** Examples of typical derivatives of SpiroOMeTAD. **c** Influence of substitution pattern on the resulting electronic and physical properties[16]. HTM = hole transport material.

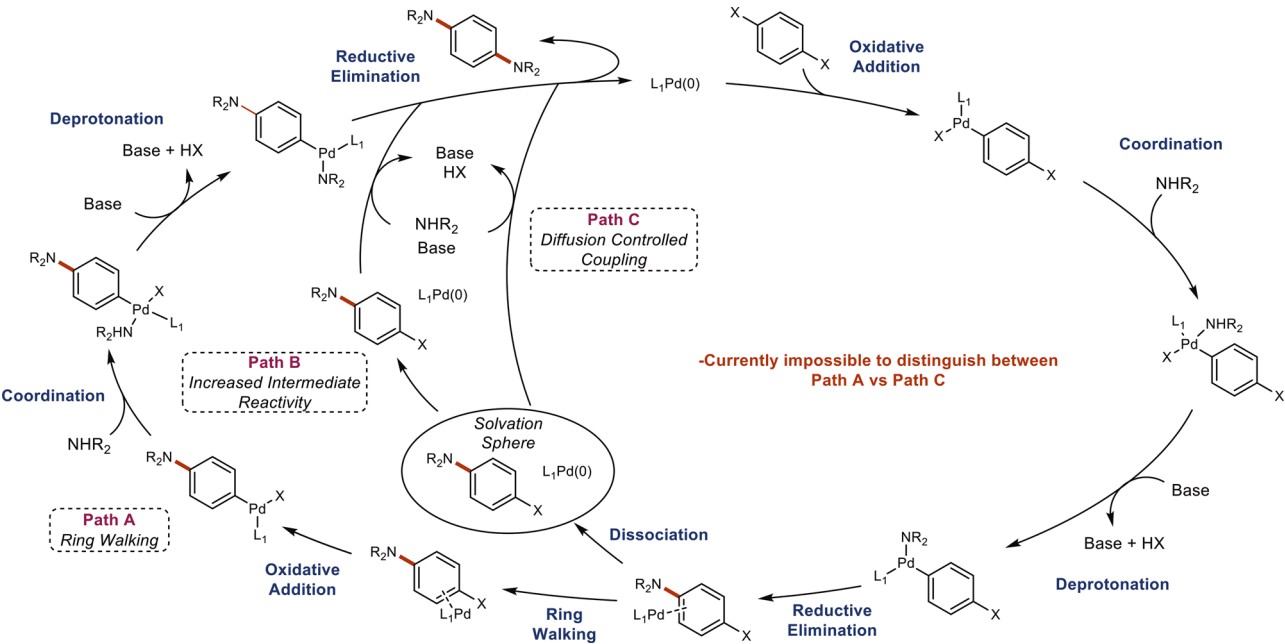

**Fig. 2 Prototypical Buchwald-Hartwig amination mechanism.** Prototypical Buchwald-Hartwig amination with ring walking in which the catalyst remains bound to the pi system throughout the cycle (path A), increased intermediate reactivity, which can often be mistaken for ring walking (path B), and diffusion-controlled coupling in which the catalyst remains within the solvation sphere of the intermediate and begins a second round of coupling (path C). Prior to this work, it was impossible to distinguish between paths A and C.

derivatives. Finally, the complexity of the model system studied enables the differentiation between authentic ring walking and diffusion-controlled coupling mediated by solvation effects.

## Results and discussion

**Model system**. The targeted reaction is difficult to monitor via standard spectroscopic techniques due to its air-sensitive nature and the number and structural similarity of relevant species: there are four potential intermediates in addition to the desired product (Fig. 1a). Consequently, kinetic studies on systems of such complexity are exceedingly rare. However, a mechanistic study on such a system would enable access to insights unattainable when studying simpler systems. Specifically, the presence of several halogens on two distinct pi systems would enable the differentiation between the mechanistic phenomenon of ring walking versus solvent mediated diffusion-controlled coupling if reliable time course data is accessible.

There are 4 potential scenarios which could be observed and differentiated by targeting the SpiroOMeTAD model system (Fig. 3):

1. Ring walking and diffusion control: In this case, one would expect the formation of the dicoupled intermediates selectively, however, the rate of product formation would not be dependent on the concentration of the intermediate(s) (Fig. 3, path 1).
2. Ring walking with no diffusion control: Selectivity would be expected for the formation of a dicoupled product with both amines on the same fluorene ring system. The rate of product formation would depend on the concentration of the dicoupled intermediate (Fig. 3, path 2).
3. No ring walking with diffusion control: A mixture of intermediates would be expected where the rates of formation of intermediates and the product do not depend on the concentration of their parent species. Crucially, no selectivity would be expected for either dicoupled product as a statistical mixture would be expected. Alternatively, a

preference for the dicoupled product on opposing fluorene ring systems could be favored due to electronic reasons[36,37]. This results from the increased electron density of the monocoupled intermediate disfavoring coupling at the remaining electrophilic site of that pi system. (Fig. 3, path 3).

4. No ring walking and no diffusion control: In such a system, all potential intermediates would be observed to build in and decay over time. A preference for the formation of the dicoupled product on opposite sides of the fluorene ring system would be expected. Moreover, the rate of formation of each intermediate and the product would depend on the concentration of their parent species (Fig. 3, path 1).

Thus, we began our study by leveraging our previously reported reaction monitoring platform to study our targeted model system. We identified four ligands commonly employed in the palladium-catalyzed synthesis of triarylamines, namely Pd(OAc)$_2$/P($t$Bu)$_3$, XantPhos Pd G4, RuPhos Pd G4, and PEPPSI-IPr (Fig. 3b)[38]. The goals of this study were to: (1) understand how ligand choice affects the evolution of intermediates throughout the course of the reaction, with a focus on ring walking behavior, (2) probe the influence of the catalyst resting state on ring walking, (3) investigate the generality of our findings on alternate polyhalogenated cores, and (4) identify conditions which could provide access to asymmetric SpiroOMeTAD derivatives.

**Ligand effects on SpiroOMeTAD synthesis**. We began our study by probing P($t$Bu)$_3$ as the ligand. The combination of Pd(OAc)$_2$/P($t$Bu)$_3$[5,10,38,39] was targeted as it is the most frequently employed catalyst system used in the synthesis of SpiroOMeTAD and its derivatives. The standard time course data of Pd(OAc)$_2$/P($t$Bu)$_3$ can be seen in Fig. 4b. Notably, the presence of only a single intermediate was observed under the reaction conditions. This intermediate was isolated, fully characterized, and identified as dicoupled intermediate (**4a**). We obtained a crystal structure of

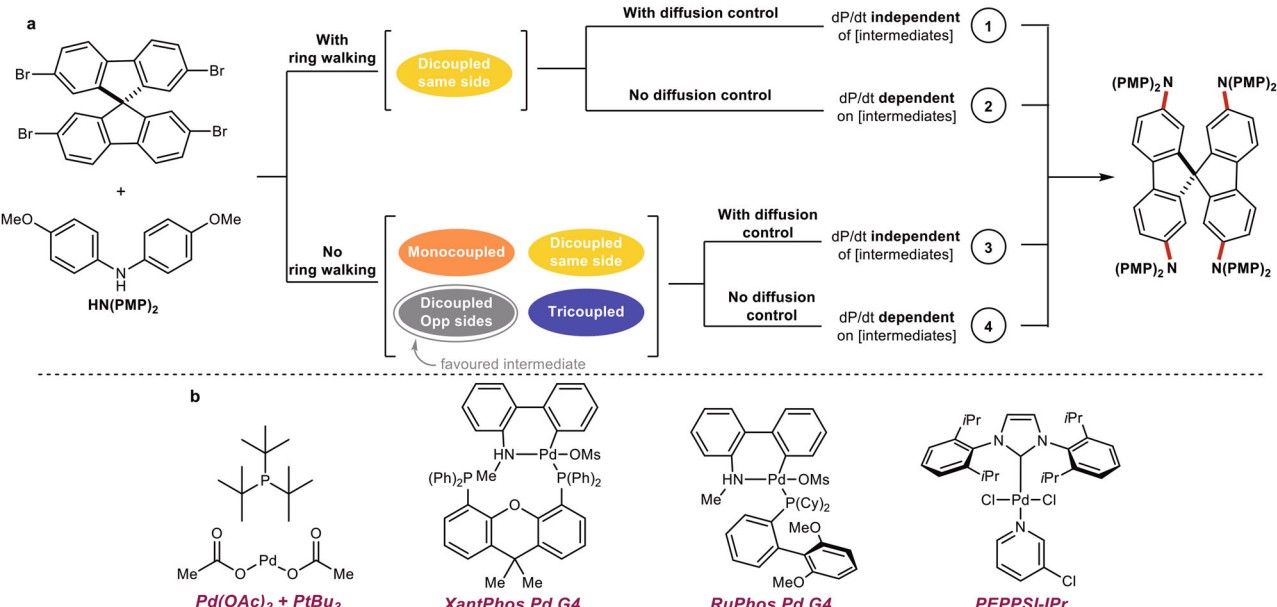

**Fig. 3 Model system and targeted catalysts. a** Potential mechanistic regimes present for the SpiroOMeTAD model system enabling differentiation between authentic ring walking versus solvent-mediated diffusion control. **b** The 4 catalyst/ligand combinations targeted for the study.

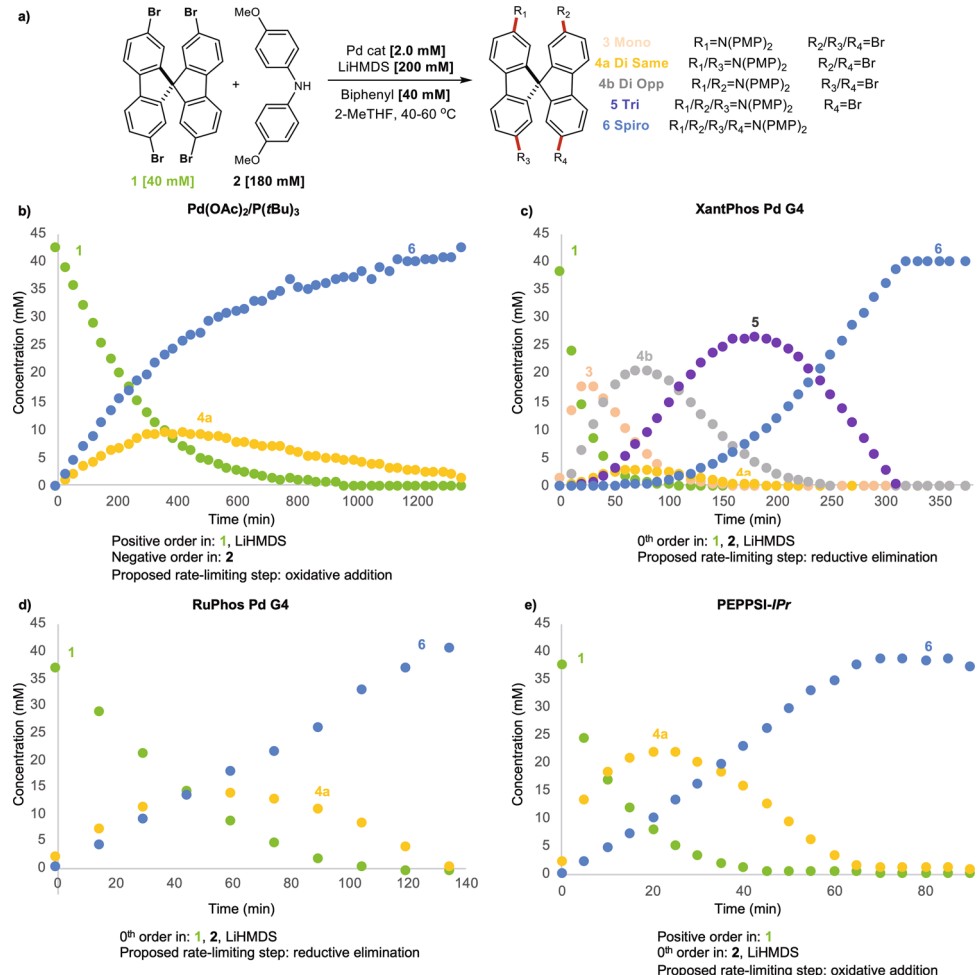

**Fig. 4 Standard time course data of targeted catalyst. a** Reaction scheme and conditions for standard condition time course reaction profiles. **b**, Standard time course for Pd(OAc)$_2$/P($t$Bu)$_3$ and summary of kinetic experiments. **c** Standard time course for XantPhos Pd G4 and summary of kinetic experiments. **d** Standard time course for RuPhos Pd G4 and summary of kinetic experiments. **e** Standard time course for PEPPSI-IPr and summary of kinetic experiments.

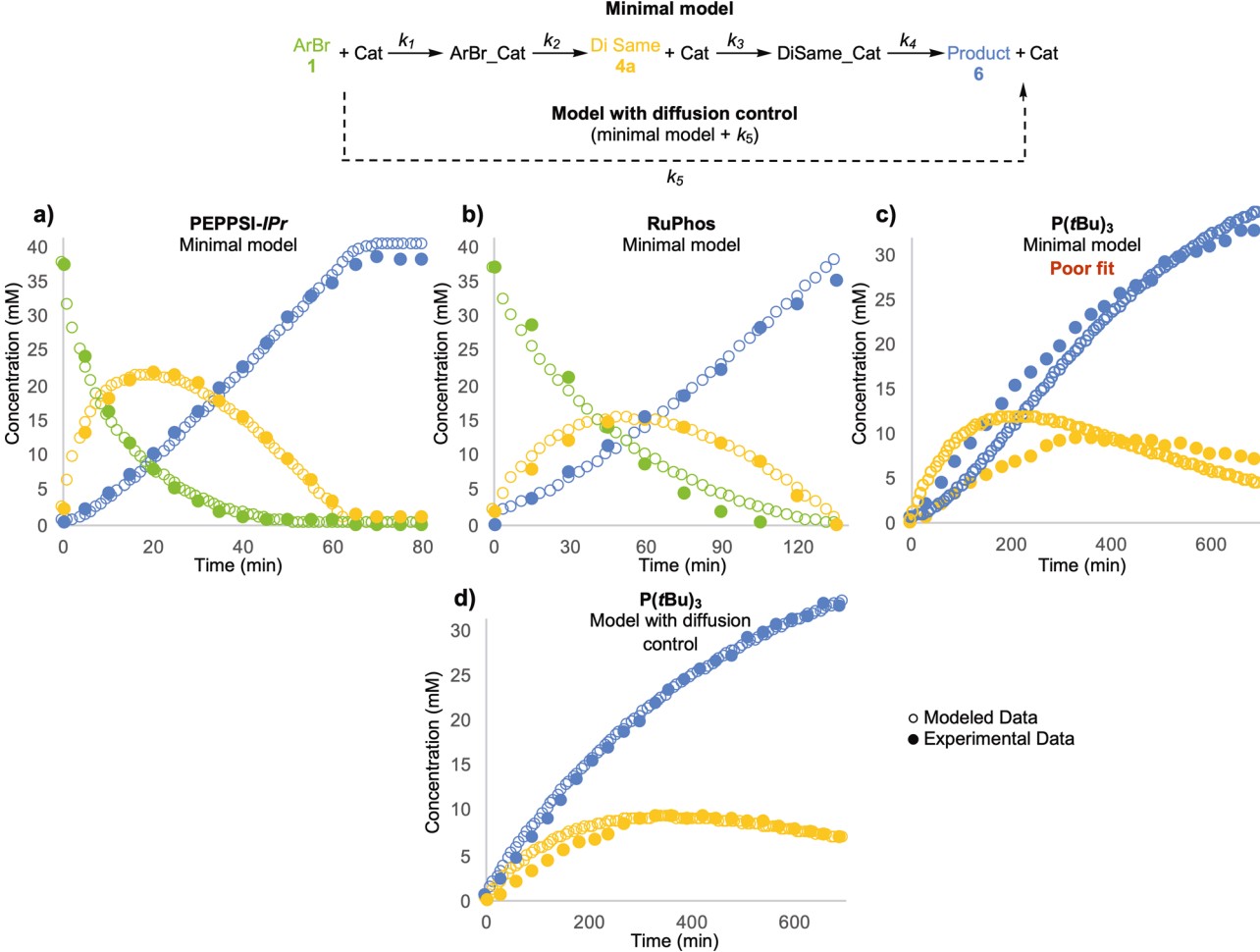

**Fig. 5 COPASI model of monodentate ligands. a** The COPASI modeled data without diffusion-controlled coupling overlaid with the acquired experimental data for PEPPSI-IPr. **b** The COPASI modeled data without diffusion-controlled coupling overlaid with the acquired experimental data for RuPhos Pd G4. **c** The COPASI modeled data without diffusion-controlled coupling overlaid with the acquired experimental data for Pd(OAc)$_2$/P($t$Bu)$_3$. **d** The COPASI modeled data with diffusion-controlled coupling overlaid with the acquired experimental data for Pd(OAc)$_2$/P($t$Bu)$_3$.

the intermediate to confirm that **4a** has both diaryl amine moieties on the same fluorene system (see supplementary Fig. 22). If solvent mediated diffusion-controlled coupling was operative, a mixture, or an outright preference for the coupling on opposing sides of the fluorene ring system would be expected due to the increased electron density disfavoring **4a** following the formation of the first C-N bond[36,37]. This result taken together with the lack of any observable monocoupled intermediate provide some of the strongest evidence to date for the mechanistic phenomenon of ring walking (Fig. 2, path A). Moreover, the ability of P($t$Bu)$_3$ to promote ring walking under BHA conditions is in line with previous observations from Suranna et al.[26]. Notably, the rate of SpiroOMeTAD (**6**) formation is greater than that of **4a** throughout the entire reaction, discussed vide infra.

A full kinetic analysis of the reaction was conducted to gain insight into the resting state of the catalyst and turnover-limiting step (see supplementary Fig. 8). Both the aryl halide (**1**) and the base were observed to have a positive order, while amine (**2**) displayed a significant negative order. The rate dependence on the aryl halide (**1**) would be indicative of a turnover-limiting oxidative addition. Hartwig and coworkers have previously observed slight positive order in strong anionic bases under such conditions but this behavior was limited to aryl chlorides[40]. The increase in steric bulk of LiHMDS compared to NaO$t$Bu would be expected to further diminish such an effect. The negative order in

amine was unexpected given the weak coordination ability of diarylamines and limited reports of amine inhibitory behavior present in the literature[41]. This result suggests that the formation of an off-cycle species between the amine and the palladium significantly impacts the overall kinetics. The inhibitory effect of **2** was further confirmed through a dosing experiment in which vastly increased rates were observed through slow dosing of the amine in the reaction system over time (see supplementary Fig. 9). We tentatively propose a turnover-limiting oxidative addition in which the behavior observed from **2** and LiHMDS arises from a complex interplay between catalyst initiation and decay. This is the subject of ongoing studies in our laboratories.

We then turned our attention to the use of a bidentate phosphine system. XantPhos was targeted as a representative ligand as it is commonly employed in the coupling of diarylamines[38] and has been reported in the synthesis of **6**[42]. The XantPhos Pd G4 precatalyst was employed to overcome irreproducible catalyst activation behavior. The time course plot can be seen in Fig. 4c. In stark contrast to P($t$Bu)$_3$, four intermediates were clearly present under the tested reaction conditions. Moreover, a clear preference for the dicoupled intermediate on opposing sides of the spirocyclic ring system (**4b**) was observed. This time course data strongly supports the inhibition of the ring walking previously observed in the P($t$Bu)$_3$ system. These results highlight the robust nature of the automated sampling system by providing reliable time course data

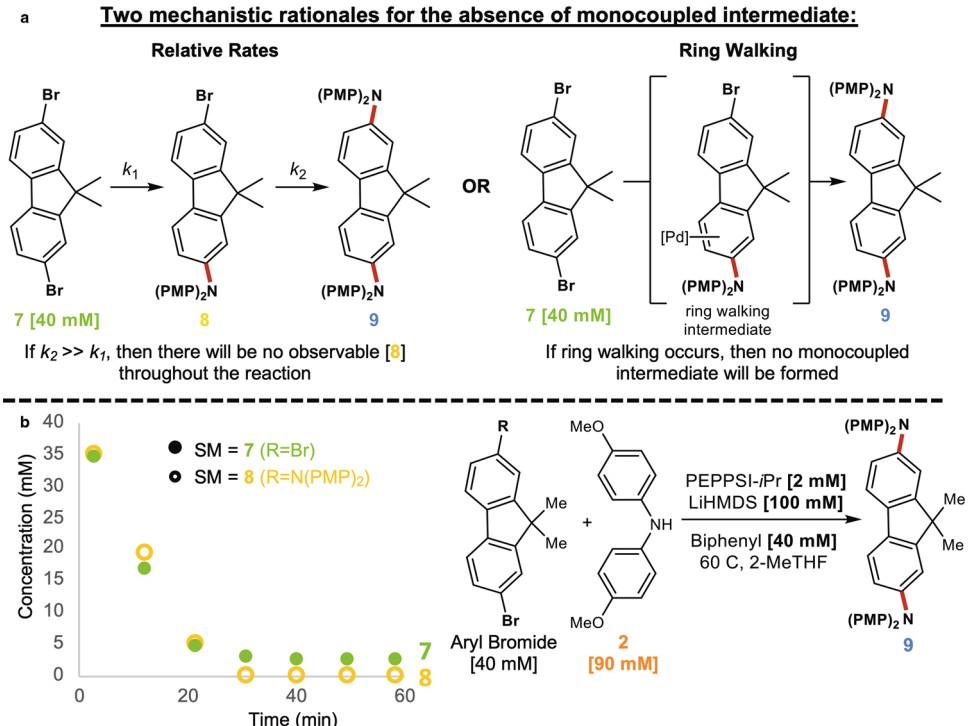

**Fig. 6 COPASI model of XantPhos Pd G4.** Overlay of the experimental and modeled time course data of the standard conditions reaction using XantPhos Pd G4 as the precatalyst.

**Fig. 7 Probing alternative mechanistic explanation. a** Depicting two mechanistic rationales for the absence of observable intermediates under some conditions. **b** Overlay of **7** vs **8** consumption under standard reaction conditions shows identical rates.

for a complex system. A full kinetic analysis of this system was then conducted (Fig. 4c and supplementary Fig. 13). A 0$^{th}$ order dependence in **1**, **2**, and LiHMDS suggests that a turnover-limiting reductive elimination is operative under the conditions studied.

Given the stark contrast between monodentate and bidentate phosphines on ring walking, we chose to probe the use of dialkylbiarylphosphines. This versatile ligand class has seen extensive use in BHA and are known to exhibit weak chelation through the lower aromatic ring of the phosphine ligand[43]. A number of dialkylbiarylphosphines have been reported in the synthesis of **6** and we chose to move forward with RuPhos Pd G4 as a precatalyst[42]. RuPhos has been developed as a robust ligand

to achieve selective couplings of secondary alkylamines and diarylamines. A standard time course plot can be seen in Fig. 4d. **4a** was the only intermediate detected and suggests ring walking is operative when using RuPhos, despite the weak chelating nature of the ligand[43]. These results are in line with results in C-C bond forming systems[44]. A full kinetic analysis again revealed a 0$^{th}$ order dependance in **1**, **2** and LiHMDS suggesting a turnover-limiting reductive elimination (see supplementary fig. 12). Such a mechanistic regime is in line with a recent report by Buchwald et al. in a simplified system[45].

Finally, we turned our attention to *N*-heterocyclic carbenes (NHC) as ligands in the current study. NHCs have been broadly

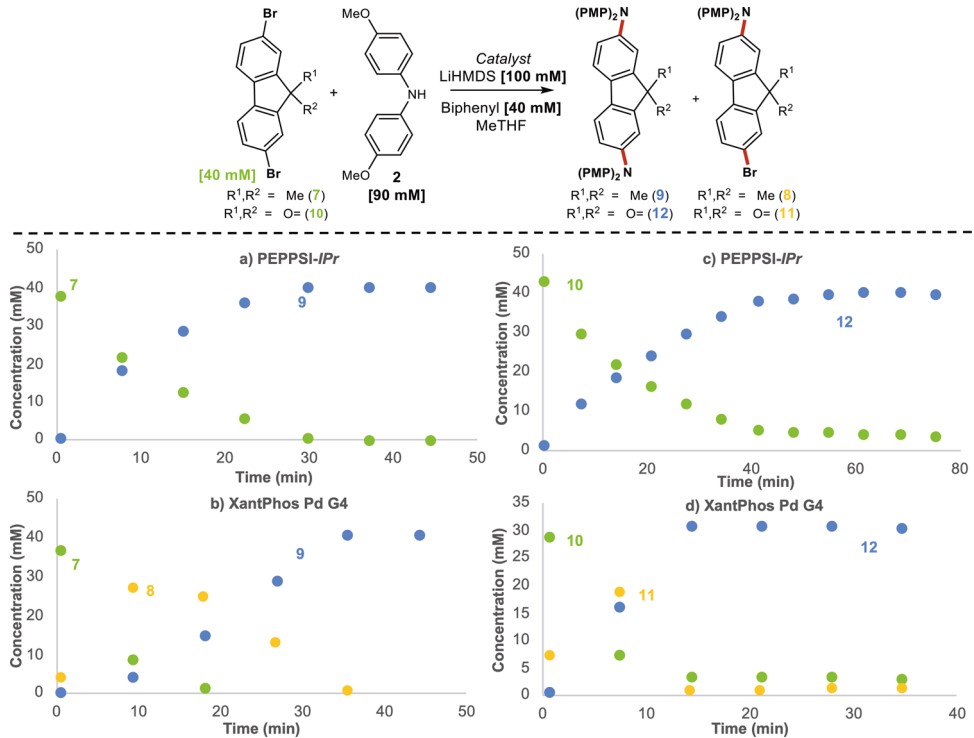

**Fig. 8 Time course data on alternative dibrominated substrates. a** Time course data for coupling of **7** with PEPPSI-IPr. **b** Time course data for coupling of **7** with XantPhos Pd G4. **c** Time course data for coupling of **10** with PEPPSI-IPr. **d** Time course data for coupling of **10** with XantPhos Pd G4.

applied in BHAs and many NHC precatalysts are readily available[46]. A NHC based palladium catalyst has been described in the synthesis of **6**, however the ligand was highly specialized and not commercially available[47]. We thus chose to move forward with PEPPSI-IPr as our model NHC system due to its commercial availability and extensive use in BHAs[46]. As seen in Fig. 4e, the standard time course plot for PEPPSI-IPr revealed a similar intermediate profile as observed with P(*t*Bu)₃ and RuPhos, supporting ring walking behavior. A full kinetic analysis of the system revealed a positive order in **1** and 0ᵗʰ order in **2** and base (Fig. 4e), although it is worth noting that significant inhibition was observed when LiHMDS was used in large excess relative to **2** (see supplementary Fig. 11). These results suggest that a turnover-limiting oxidative addition is operative under these conditions.

**Analysis of kinetic data**. Analyzing the kinetic data globally has interesting implications for ring walking behavior (Fig. 4b-e). The resting state of the catalyst seems to have little effect on its propensity to ring walk. This is clearly observed when comparing RuPhos and XantPhos. In both cases, a turnover-limiting reductive elimination is observed while only RuPhos displays ring walking behavior. Instead of the catalyst resting state, the use of monodentate versus bidentate ligands was observed to have a drastic effect on ring walking behavior in these systems. Moreover, the mechanistic studies provided herein highlight an opportunity to further corroborate ring walking in new reaction systems. Studies of ring walking have largely focused on C-C bond forming systems in which the catalyst pi complex eludes direct observation due to its fleeting nature[28–30]. In contrast, the results obtained with PEPPSI-IPr and P(*t*Bu)₃ suggest a Pd(0) resting state which could allow for direct observation.

**COPASI modeling**. The time course profile for each ligand was modeled using COPASI to gain insight about the observed intermediate distributions. Best practice with respect to reaction modeling suggests the use of the simplest possible model to explain the observed behavior to avoid obtaining overdetermined results. We began with a simple model wherein starting material and catalyst are converted directly to product. Additional steps were added until the model fit the experimental data. The minimal model obtained for PEPPSI-IPr (Fig. 5a) and RuPhos (Fig. 5b) required four steps, including the formation of intermediary aryl halide-catalyst complexes. This model did not include the formation of the mono or tri coupled intermediates. The magnitude of the fitted k values was similar for each step, demonstrating that each step in the model is kinetically relevant.

When the minimal model was applied to the P(*t*Bu)₃ data, there was poor agreement between the modeled and the fitted data (Fig. 5c). Without diffusion-controlled coupling, one would expect an accumulation of **4a** early in the reaction followed by its decay and the formation of **6**, as is observed in the case of PEPPSI-IPr. In contrast, the rate of formation of **6** was significantly greater than that of **4a** when using P(*t*Bu)₃. The poor overlap observed early on in the reaction of P(*t*Bu)₃ suggests this model is insufficient for the system at hand. We hypothesized that a diffusion-controlled coupling could explain this behavior where, upon formation of **4a**, the palladium remains within the solvation sphere of the intermediate and undergoes a second round of couplings forming **6** directly (Fig. 2, path C). This behavior would explain why the rate of formation of **6** can be greater than or equal to the rate of formation of **4a**. To substantiate this claim, our model was updated to allow for this diffusion-controlled coupling behavior (Fig. 5d) and a significantly improved overlay was observed suggesting both ring walking and diffusion controlled coupling are operative for this ligand (Fig. 3a, path 1).

XantPhos required a more complex model that included the formation of all intermediates and aryl halide-catalyst complexes (Fig. 6). This twelve-step model did not account for diffusion control or ring walking, suggesting that neither of these phenomena are operative in the XantPhos system. The magnitude

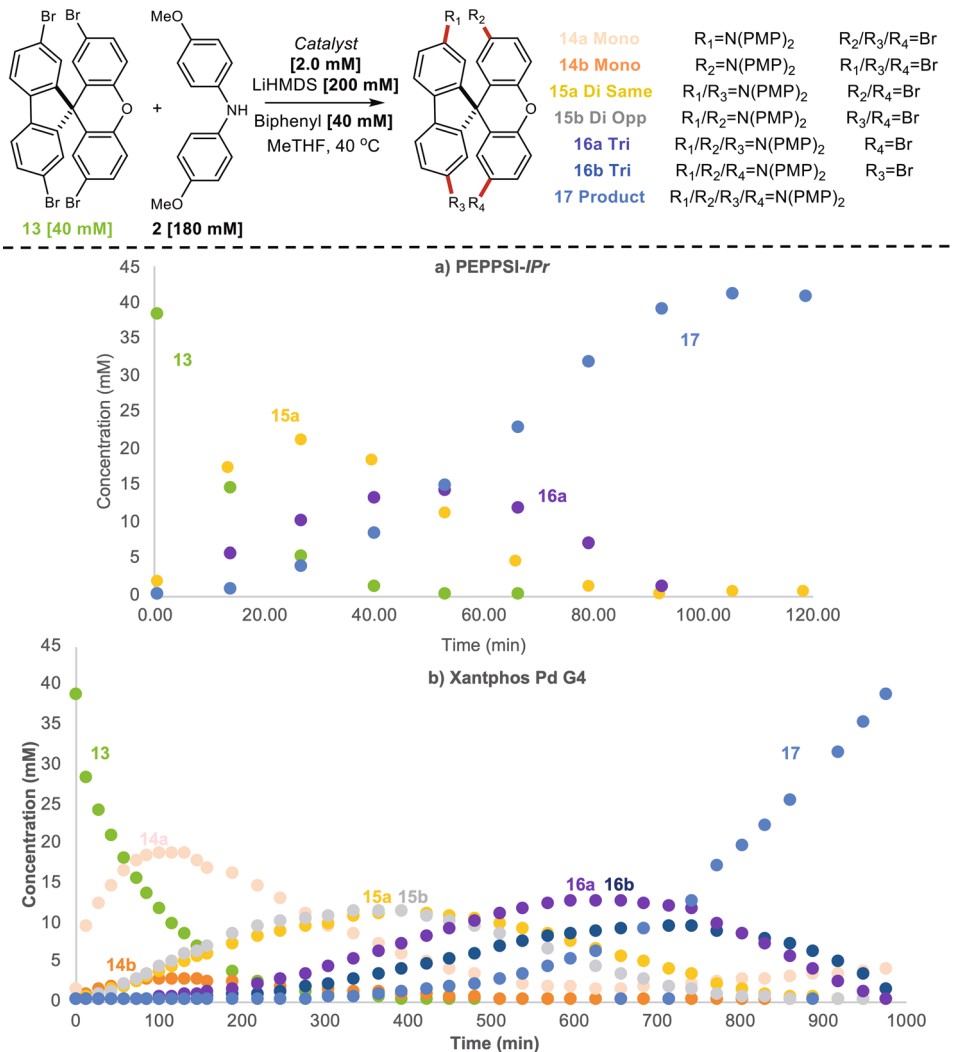

**Fig. 9 Time course data on alternative tetrabrominated substrate. a** Time course data for coupling of **13** with PEPPSI-IPr. **b** Time course data for coupling of **13** with Xantphos Pd G4.

of the k values was similar for each step apart from the steps involving the formation of the di coupled intermediates. The k values $k_3$ and $k_4$ represent the affinity of palladium to bind the monocoupled intermediate either on the same ring ($k_3$) or the opposite ring ($k_4$). If irreversible palladium oxidative addition is assumed, as in this model, then the ratio of $k_4$:$k_3$ should determine the selectivity between **4a** and **4b**. The ratio of $k_4$:$k_3$ is ~10:1 which matches the [**4b**]:[**4a**] observed in the experimental data. Thus, this model demonstrates that the regioselectivity in the XantPhos system is due to a kinetic selectivity.

**Alternative mechanistic interpretation**. The ability of intermediates to display increased reactivity is a well known phenomenon that can lead to false positives when developing ring walking catalysts[31]. A possible alternative explanation for the behavior observed under the systems studied could be a consequence of intermediates which display a significant increase in reactivity relative to their parent aryl halide starting material (Fig. 2, path B; Fig. 7a). This could explain the observed preference for the formation of **4a** as well as the inability to detect mono or tricoupled intermediates with certain catalysts. To probe this possibility, we isolated monoaminated intermediate **8** and studied its reactivity under standard reaction conditions with PEPPSI-IPr. Notably, this experiment in fact biases **8** to display

greater reactivity than it would otherwise under normal reaction conditions as a result of its greater concentration (ie: 40 mM) compared to standard conditions with **7** (ie: not detected). An overlay of the decay profile of **7** vs **8** can be seen in Fig. 7b. Notably, the rate of consumption is identical for both species. This result clearly shows that the intermediate does not have a pronounced increase in reactivity, further bolstering our conclusion that authentic ring walking behavior is being observed in the systems studied.

**Generality of observed ligand effects**. To support the generality of the results observed in the present study, experiments on alternative polyhalogenated substrates were undertaken. PEPPSI-IPr and XantPhos Pd G4 were chosen as representative precatalysts in the amination of **7**, **10**, and **13** (Figs. 8 and 9). Results using substrate **7** supported ring walking behavior similar to the SpiroOMeTAD system (Fig. 8a, b). Importantly, this behavior was also observed with the introduction of an electron-withdrawing group such as the ketone in **10** (Fig. 8c, d). Electron-poor substrates have a lower propensity for efficient ring walking, and reactions with these substrates remain a challenge within the field of CTP[48,49]. Finally, we probed the use of tetrabrominated spirocycle **13** (Fig. 9). Results on this system also corroborate our initial findings but display some unforeseen complexities. In the

**Fig. 10 Synthesis and derivatization of asymmetric SpiroOMeTAD derivatives.** Desymmetrization of **1** conditions: **1** (0.40 mmol), HN(PMP)$_2$ (0.55 mmol), LiHMDS (0.60 mmol), PEPPSI-IPr (2.5 mol%), MeTHF (0.04 M), 60 °C, 3 h. Conditions (**a–d**) **4a** (0.30 mmol), K$_3$PO$_4$ (8.0 equiv), Water:THF 1:4 (0.075 M), 50 °C, 16 h. **a** 4.0 equiv. 3-thienylboronic acid. **b** 4.0 equiv. 4-butylphenylboronic acid. **c** 4.0 equiv. 4-biphenylboronic acid. **d** 4.0 equiv. 3-furanylboronic acid.

case of PEPPSI-IPr, a dicoupled intermediate (**15a**), a tricoupled intermediate (**16a**), and the product were all observed under the reaction conditions (Fig. 9a). In contrast, XantPhos produced a complex mixture of intermediates and product due to the presence of regioisomers, indicating the inhibition of ring walking (Fig. 9b). The intermediate profile observed with PEPPSI-IPr does support ring walking behavior however the presence of the oxygen on the xanthene core appears to inhibit ring walking resulting in the formation of **16a**.

**Accessing asymmetric derivatives of SpiroOMeTAD**. Finally, we turned our attention to accessing asymmetric SpiroOMeTAD derivatives from **1** by leveraging the knowledge obtained throughout this study. We targeted the use of PEPPSI-IPr as its lack of diffusion-controlled coupling behavior resulted in significant differentiation between the formation of **4a** and **6**. Reducing the amount of **2** and LiHMDS enabled us to access **4a** with a 61% yield based on remaining starting material (Fig. 10). **4a** was then successfully derivatized using a Suzuki-Miyaura cross coupling. This provides a straightforward methodology to access asymmetric SpiroOMeTAD derivatives enabled through an in-depth understanding of ligand effects in such systems. Given the widespread use of polyhalogenated precursors in BHAs, we expect the results presented herein to have a significant impact on enabling chemists to access a broader chemical space when undergoing library syntheses[23].

A mechanistic analysis of ligand effects on the evolution of intermediates during SpiroOMeTAD synthesis was conducted with a focus on understanding ring walking behavior. Our results suggest that the resting state of the catalyst does little to promote

or inhibit ring walking. Instead, monodentate ligands were observed to promote ring walking while the bidentate ligand tested inhibited ring walking. These results were also observed to be general when applied to alternative polyhalogenated systems. Furthermore, the ability to experimentally probe ring walking behavior and delineate its impact compared with solvent-mediated diffusion control was reported. This highlights how careful substrate choice coupled with a reaction monitoring technique amenable to complex systems can provide mechanistic insights which escape simpler model reactions typically targeted for mechanistic investigations. Leveraging advanced analytical technology to quantify time course profiles of multicomponent mixtures is a key enabling technology. Finally, the knowledge obtained in this study was leveraged to access asymmetric derivatives of SpiroOMeTAD, highlighting the ease with which libraries of these compounds can be accessed for screening.

## Methods

**General procedure for P(tBu)$_3$/Pd(OAc)$_2$ reaction monitoring**. An oven-dried 4 dram vial was brought into the glovebox where **1** (0.253 g, 0.400 mmol), **2** (0.413 g, 1.80 mmol), and LiHMDS (0.335 g, 2.00 mmol) were added along with a magnetic stir bar. This was followed by the addition of THF (9.0 mL) and toluene (0.80 mL). The vial was sealed with a pierced rubber septa through which the EasySampler probe was inserted. The reaction mixture was heated to 60 °C. Meanwhile, an oven dried 1.5 mL vial was brought into the glovebox where Pd(OAc)$_2$ (8.98 mg, 4.00×10$^{-2}$ mmol) and P(tBu)$_3$ (65 uL of a 1.24 M sol'n in toluene) were added and diluted with toluene (0.335 mL). This solution which is initially red is stirred until it becomes colorless, upon which 0.20 mL of the solution is added to the 4 dram reaction vial to start the reaction.

**General procedure for bench stable catalyst reaction monitoring**. To an oven-dried 15 mL two-neck round bottom flask in the glovebox was added LiHMDS (334 mg, 2.00 mmol) and a magnetic stir bar. The flask was sealed with two rubber

septa, removed from the glovebox, and put under argon on a Schlenk line. Under positive pressure, one septum was removed and **1** (259.2, 0.40 mmol), **2** (412 mg, 1.80 mmol), and biphenyl (61.7 mg, 0.400 mmol) were added to the reaction flask. The flask was resealed with a septum and the system was evacuated for five minutes and backfilled with argon three times. Then, under a high flow of argon one septum was removed and the EasySampler probe was inserted into one neck of the round bottom flask. 2-MeTHF (9.75 mL) was then added to the reaction flask upon which the flask was heated to 60 °C. Meanwhile, a precatalyst solution was prepared by adding the appropriate precatalyst ($3.0{\times}10^{-2}$ mmol) to an oven-dried 1.5 mL vial. The vial was evacuated for five minutes and backfilled with Ar three times before 0.75 mL of 2-MeTHF was added to the vial. The reaction was initiated by transferring 0.50 mL of the precatalyst solution to the reaction flask. Note: reaction monitoring involving **7**, **10**, and **13** were conducted with this same procedure based on a 0.40 mmol scale for the aryl bromide. **2** and LiHMDS were employed in 1.25 equivalents and 1.50 equivalents per aryl halide bond.

## Data availability

Characterization of the products, experimental procedures, detailed kinetic data, and COPASI modeling data are available in the Supplementary Information. Crystallographic data for structure **4a** reported in this article has been deposited at the Cambridge Data Center under deposition number 2102751 (https://www.ccdc.cam.ac.uk/structures/).

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

## Acknowledgements

We gratefully acknowledge Mettler-Toledo Autochem for their generous donation of process analytical equipment (EasyMax and EasySampler) to J.E.H. Financial support was provided by Natural Resources Canada (EIP2-MAT-001) to C.P.B. and J.E.H. Additional financial support was provided by the University of British Columbia, the Canada Foundation for Innovation (CFI-35883), the Natural Science and Engineering Research Council of Canada (RGPIN-2021-03168) and the Defense Advanced Research Projects Agency (DARPA) for funding this project under the Accelerated Molecular Discovery Program (Cooperative Agreement No. HR00111920027, dated August 1, 2019). C.P.B. is grateful to the Canadian Institute for Advanced Research (BSE-BERL-162173) for financial support. We further wish to acknowledge Paloma Prieto for her assistance with the manuscript.

## Author contributions

M.C.D. performed the kinetic experiments with XantPhos Pd G4, RuPhos Pd G4, and PEPPSI-IPr. T.C.M. and K.L. performed the $P(tBu)_3$ kinetic experiments. J.S.D. and M.C.D. ran the derivatizations and undertook the characterizations. C.B.P., J.E.H., J.S.D, and T.C.M. helped devise the project. J.S.D, M.C.D., T.C.M., C.B.P. and J.E.H. prepared the manuscript and supplementary information.

## Competing interests

The authors declare no competing interests.
