## [Peer Review File · Nature Communications]

REVIEWER COMMENTS

Reviewer #1 (Remarks to the Author):

The submitted manuscript by Berlinguette and Hein describes a series of studies on Hartwig-Buchwald Amination reactions to synthesize SpiroOMeTAD derivatives. While the work is interesting, it is too preliminary for publication currently. Moreover, given the narrow substrate scope and ligands investigated, even when fully fleshed out, I think this work is more appropriate for a specialized journal.

I have several significant concerns with the kinetic data and its interpretation:

- Only one set of data was duplicated for each catalyst and ALL remaining experiments (most of the SI and all the conclusions in the paper) are based off a single set of data. This is not good practice and should not go into the literature this way. Every data should be (at least) duplicated.
- The authors have not characterized ANY of the resting states and it's not clear why. These catalysts should be amenable to ³¹P NMR spectroscopic analysis with the phosphine-based ligands. Instead, the authors make assumptions IN EVERY CASE about what the resting state "must" be.
- The authors assume reaction orders based (literally) on two data points. They have the "standard conditions" and one other concentration (either higher or lower). This is also not good practice. Of course, when you fit two points, you will always get a line (zero order, first order). But you could end up mis-assigning these orders or missing important information (e.g., saturation behavior, 2nd order, inhibition).
- I encourage the authors to carefully read the Hartwig/Buchwald paper in JACS in 2006 (pg 3584) to understand more the complexities that are likely in these rate studies and why careful, controlled studies are important, in addition to having a clear understanding of the actual resting states.
- Looking at the SI ... it looks like many of these reactions do not go to full conversion of 1. What is going on in those cases? Like it levels off at 50% or less.
- Table 1 really highlights my misgivings. The entire paper is premised on a clear understanding of the kinetic data but the rxn orders are only based on two points (I'm not even clear if the authors fit these decay profiles or just eye-balled it), and the turnover limiting steps are potentially wrong b/c they haven't identified the catalyst resting states. Basically, I'm not confident in the first four columns of data. The only thing I think they might have done correctly is assign whether it's ring-walking or not.

Other thoughts:

1. Overall, each reaction uses superstoichiometric amounts of the amine coupling partner. Several studies (Larrosa et. al. *Org Lett* 2011, 13, 146 cited by these authors; Groombridge et. al. *Chem. Comm.* 2015, 51, 3832), use substoichiometric quantities of coupling partner to reduce false positives for ring-walking hits. Can the authors observe similar reactivity trends when using substoichiometric amine?

2. The authors comment on the difference between mono/bi dentate phosphines but only briefly mention differences between alkyl/aryl phosphines. It might be interesting to see the difference between mono/bi dentate with alkyl/aryl phosphines. For example, P(Ph)₃ vs P(tBu)₃ or, for Buchwald ligands DavePhos or Jackie Phos vs RuPhos.

3. The authors provide evidence for the fact that mono- vs bi-dentate ligands affect ring-walking, but don't rationalize these results. Can the authors provide some insight for why the mono-dentate ligands don't dissociate? (sterics seems like the easy answer) And why the NHC ligand is more comparable to the mono-dentate phosphines?

4. Substrate scope – I understand that they chose the scope of SpiroTADs based on existing hole-transporting materials. However, I think more relevant/comparable substrates could have been selected to prove the generalizability of the ring-walking findings. For example, instead of using a ketone-substituted spirocycle (Csp2) for an "electron deficient" substrate, they could have chosen a di-fluorinated spirocycle (Csp3), which is used as a monomer for conjugated polymers. The authors also do not report an asymmetrically substituted derivative of substrate 13, which was pointed out in their introduction. Also, the authors cite the fact that the electron-rich amine should theoretically deactivate the spirocyclic core to OA as additional evidence for ring-walking. Given that the amine can "inhibit" reaction rate for coupling performed with P(tBu)₃ I'm curious if/how the electronics of the amine affect the rate determining step and/or ring-walking

5. Probing diffusion vs ring-walking: The authors are missing some examples from the conjugated polymer literature using capping agents and/or catalyst traps to differentiate between diffusion based coupling and ring walking by analyzing polymer end-groups for catalyst-transfer polymerization. For example, please see:

- a. Koeckelberghs and coworkers Polym. Chem. 2011, 49, 5339.**
- b. McNeil and coworkers JACS 2018, 140, 7846.**

6. This acronym is never spelled out: SpiroOMeTAD

7. In Figure 2A, you have what should be a Pd(III) intermediate (after deprotonation) or its anionic Pd(II) or maybe it doesn't exist? See Hartwig/Buchwald paper mentioned above.

Reviewer #3 (Remarks to the Author):

In this manuscript, the authors investigate the ability of a Pd catalyst to "walk" along the backbone of an aromatic moiety after a successful C-N coupling to carry out a second C-N coupling. This ring walk has been concluded and studied earlier and the authors now take a look at the particular case of tetrabrominated 9,9'-spirobifluorenes. The goals of this work are to provide further support of a ring walk, to investigate its ligand dependency, and to apply it in the synthesis of unsymmetric SpiroOMeTAD derivatives. The latter are of importance for functional materials research and hard to synthesize otherwise.

The authors discuss that for a double C-N coupling at a 1,4-dibromoaryl unit (and analogous also at a 4,4'-dibromobiphenyl moiety) three pathways A-C can be followed: A) via the ring walk, B) via a solvation sphere that keeps the catalyst close, and C) under diffusion control. The latter refers to a complete diffusion and reapproach of the catalyst before the second (third, fourth) C-N coupling. The authors argue that the observation of only the intermediate at which both C-N couplings had occurred at the same biaryl (fluorene) moiety supports this ring walk scenario. Indeed, having the second C-N coupling occur at the same biaryl system would be in agreement with the catalyst not detaching from this unit. The second oxidative addition at the aminobiaryl unit would be otherwise electronically disfavored and slower than a second oxidative addition at

the remaining, more electron-poor dibromobiaryl unit. However, the ring walk is only one possible explanation and there may be other causes for the observed preferences. The authors have recorded several sets of kinetic data with four different ligands that show varying behavior: P(tBu)₃ and RuPhos show only intermediate 4a (both C-N couplings at the same biaryl unit) whereas Xanthphos and a PEPPSI ligand show a mixture of all possible mono, bi, and trisubstituted intermediates. In addition, for P(tBu)₃ the formation of 4a is slower than the product formation, which indicates two product forming pathways. A COPASI simulation using a model involving all potential intermediates and an addition equilibrium bypassing intermediate 4a led to a good overlay of the curves. For the other ligands modeling this bypass was not necessary to achieve a good overlay. The catalyst system providing the highest amount of 4a as intermediate is also used to prepare the unsymmetric products 18–21, which underlines the synthetic usefulness for making unsymmetric SpiroOMeTAD units.

I agree that the new conditions for the synthesis of unsymmetric SpiroOMeTAD derivatives using the PEPPSI catalyst greatly facilitates the synthesis of such compounds. However, whether such unsymmetric compounds have improved redox/hole transport properties for applications in materials still remains to be shown. Regarding Hartwig-Buchwald couplings in general, the insight gained from this study is only of moderate interest, because the ring walk has been discussed earlier for such C-N couplings. Even for 4,4'-dibromofluorenes and a Pd-P(tBu)₃ catalyst, this ring walk has already been investigated (see Ref.: 26, DOI: 10.1039/c8py01646a). Therefore, I see this study as an expansion on earlier works with relevance to the particular substrate class of SpiroOMeTADs, but it is not of greater relevance.

I further have my reservations regarding the kinetic experiments and conclusions drawn from them. The data may be in agreement with the proposed ring walk, but they may also have been overinterpreted. For example, the model used in the COPASI simulations is still quite complex. It contains numerous rate constants that would likely show linear dependencies if they were estimated using the limited acquired data. Moreover, the model used for the simulation of the Pd-P(tBu)₃ system is a complex variant of a simple two-path system, which could already be sufficient to simulate the observed curves. path 1: substrate  intermediate 4a  product; path 2: substrate  product. On the other hand, the formation of 6 involves four C-N couplings, each proceeding via multiple individual steps, which makes a correct simulation of the system highly challenging. The available data may therefore not be sufficient to draw reliable conclusions regarding the potential ring walk. The kinetic preference for the formation of 4a over other intermediates that is observed with certain ligands could be caused by other effects. It is not a direct proof of the ring walk. The Supporting Information is detailed, but it contains inconsistencies and seems to be a preliminary version (still parts highlighted in yellow). In addition, I have minor remarks (see below).

Overall, I do not recommend publication in Nature Communications. Publication may be possible after revision in a more specialized journal such as Chemistry – A European Journal, The Journal of Organic Chemistry, or ACS Catalysis.

Additional notes and remarks:

- 1) The experiment design and instrumental setup for recording the kinetics of Hartwig-Buchwald aminations has been described by the authors in an earlier publication (Ref. 27).
- 2) Page 3, Fig. 2a: deprotonation should release HBaseX and the resulting intermediate should no longer have X as a ligand.
- 3) Page 9, line 200–201 and Fig. 8, and SI, Suppl. Scheme 2: Here, “diffusion control” implies that the dissociation of Pd(0) and re-association is very rapid (diffusion limit), and that reductive elimination is very rapid [with P(tBu)₃]. Hence, k₁₁ corresponds to the next rate-limiting oxidative addition without ring walk, correct? This may not be easy difficult to follow and could be explained in a bit more detail.

- 4) Page 10, Fig 9: The results only show that, with the PEPPSI catalyst, the second C-N coupling is much faster than the first one (and vice versa with the XantPhos catalyst). Concluding a presence or absence of a ring walk from this change in rates may be overinterpretation. Also: the green label "7" from Fig. 9a got misplaced into Fig 9b.
- 5) Page 11, Fig 10 top: Compounds 16a and 16b should not have an identical substitution pattern. Maybe change to 16b: R1/R2/R4 = N(PMP)₂; R3 = Br?
- 6) Page 15, Ref. 28: The citation is missing (journal, volume, page numbers)
- 7) Supporting Information (general): The SI still contains several (cross)references to Schemes/Figures highlighted in yellow that require updating. This gives the SI an unfinished appearance.
- 8) Supporting Information (general): The concentrations have been determined only indirectly (using solver). I would recommend using calibration curves instead, since the intermediates 4a,b,5 can be prepared and isolated individually (in addition to the substrates and the product).
- 9) Supporting Information (general): The full characterization data (including IR, fragmentation MS, melting point) should be provided for all new compounds.
- 10) Supporting Information, Fig 3: The experiments with P(tBu)₃ do not seem to be very reproducible, which may render the conclusions drawn from the experiments with this ligand false.
- 11) Supporting Information, Fig 6: As with P(tBu)₃, the experiments do not seem to be 100% reproducible either. The curves for [2] show different start concentrations [2]_{t=0} and a different slope.
- 12) Supporting Information, Fig 9–11: How can the start concentration of 1 ([1]_{t=0}) be identical if the description states that the concentration was reduced (triangles vs circles)? Are these the correct curves?
- 13) Supporting Information, Scheme 1: Since the equilibria are connected, compound 2 should appear on top of the second arrow (as an additive). It is not produced in the first equilibrium. The same applies to the third row.
- 14) All raw data should be provided in form of tables in the SI or as csv files. The COPASI files used for simulation could be provided as well.

Reviewer #2 (Remarks to the Author):

See attached documents.

Recommendation: Accept with minor revisions

This is a solid and well-designed kinetic study into the effect of catalyst speciation on Pd ring-walking in BHA reactions. The authors have demonstrated the unique ability to distinguish between two pathways which are nearly kinetically identical. In doing so they have revealed interesting mechanistic insight into the relationship between catalyst/ligand structure and substrate interaction. In a critical step which is often missing from mechanistic investigations, they have then used this knowledge to improve upon the original process and enable selective access to previously inaccessible asymmetric products. The substrates also have their own significance to the discipline of materials science. This report will thus be of interest to scientists from a range of disciplines and is strongly worthy of publication.

The authors primary conclusions are supported with well-designed experimental evidence and chemical reasoning, with which I am in agreement. No further experimental work is needed. The following points should be considered such that the manuscript might meet a high level of scrutiny. None of these points will change the broader conclusions but will serve to help clarify and demonstrate the principles of rigorous kinetic investigation to any future readers. I offer these comments in good faith and welcome dialogue with the authors on any of these points if I have misinterpreted their results.

I strongly recommend that the authors address points # 5, 6, 7, 10, 11, 12, 18, and 19 before this manuscript is published. The remaining points are left to author and editor discretion.

General Editorial Comments

1. In Fig 1a, at first glance I confused the right-pointing arrows with retrosynthesis arrows. Upon closer inspection the intended meaning was clear, but this may confuse other readers too.
2. The first paragraph in section "Ligand effects on SpiroOMeTAD synthesis" is a giant wall of text and is not the most pleasant to read. Can it be broken into two smaller paragraphs?
3. In Fig.10a I think some of the "R" group assignments in the legend need to be fixed. Currently it appears that **14a** and **14b** are the same, and that **16a** and **16b** are the same.
4. Fig. S3 is missing description of the initial conditions, which are present in Figs. S4, S5, & S6 for the other pre-catalysts.
5. I strongly suggest including a supplementary data file including all of the raw concentration vs time data that was used to construct the figures shown in the manuscript and SI. Sharing this data demonstrates good faith in transparency for kinetic analysis and makes the job of any reviewers or future interested parties much easier.

Comments on Kinetic Analysis

6. What is the basis for the proposed positive order in LiHMDS for the P(tBu)₃ system? In figure S7, the "Increased LiHMDS" reaction can be seen to consume **1** faster, but formation of **4a** and **6** are actually both *slower* than under standard conditions! This suggests that LiHMDS does not have a positive order on the product forming reaction, but is instead causing degradation of **1**. In fact, at the end of this reaction there is only *ca.* 17mM **6** and 12mM **4a**, leaving a missing balance of 9mM, or ~24% of the initial loading of **1**. It is dangerous to make conclusions on kinetics experiments with only 76% of mass balance accounted for.

At the end of the data for this reaction the conversion of **4a** to **6** has also nearly stopped. Is this because the increased LiHMDS is also decomposing **2**, or is it affecting catalyst activity?

7. In Fig. S7, what is going on with mass balance for the "Reduced **2**" experiment? At the last data point there is *ca.* 28mM **6** and 12.5mM **4a**. Should this then require $(2 \times 12.5) + (4 \times 28) = 137\text{mM}$ of **2** having been present at the start? But the legend for this figure says that experiment only got $[\mathbf{2}]_0 = 90\text{mM}$?

8. In Figs. S9, S10, and S11, in the plots of **[1]** vs time, for the $[1]_0=20\text{mM}$ reaction, the trends of **[1]** itself have been concentration-adjusted up by 20mM so that they overlay with the initial condition of the other three experiments. I believe this was done to clearly demonstrate the comparison of conversion rate between these conditions. This is OK, but there needs to be a very clear note in the figure caption describing this manipulation and that these triangle data points are not the true values of $[1]_t$. Otherwise it can lead to confusion for someone less well versed in this type of analysis, who may interpret these plots to indicate that the reaction stalls with 20mM **1** remaining.
9. In Fig S9 (PEPPSI) the caption indicates that the reduced **2** experiment also had reduced LiHMDS (90mM and 100mM, respectively). However, this is not the case for the reduced **2** experiments for any of the other catalyst systems. Is this a typo in one or more of the figure captions, or is it how the experiments were actually performed? If it is real, then why was this change made for PEPPSI but not the other catalysts? I suspect it is because "significant inhibition was observed when LiHMDS was used in large excess relative to **2**." If so, please see my next point.
10. In the manuscript it is mentioned in two places that the PEPPSI system experiences inhabitation by LiHMDS, with a reference made to the SI. However, I was not able to find the data supporting this statement in the SI. Fig S9 depicts zero-order kinetics in LiHMDS, and I cannot find any kinetic results elsewhere that describe the behavior if LiHMDS is used in significant excess in the PEPPSI system. Perhaps this was accidentally left out of the SI?

Comments on COPASI Modelling

11. The parameters used to fit the two models to the plots in fig S14 should be reported in the SI in addition to the structure of the model. Standard practice for any scientific report is to include the necessary information for the experiments to be repeated & verified by a third party. If a kinetic model is used in an investigation, *especially* if one of the major conclusions rests on results of that model, then the full details of that model must be made available. At a minimum this includes 1) the structure of the model, 2) the model parameters, and 3) fitting results of that model to relevant experimental data. Ideally this should also include some limited demonstration of goodness-of-fit / confidence statistics (or similar).
12. The COPASI model used to fit this data is much more complex than it needs to be (vide infra). Based on the kinetic conclusions in Figs. S7, S9, S10, and S11, the majority of steps depicted in scheme S1 are not kinetically relevant and estimates of their kinetic constants from the standard reaction data will not be meaningful. I suspect that the model as applied to the data in fig. S14 is highly underdetermined with significant errors/uncertainties $\gg 100\%$ associated with most of the parameter estimations. This is still an OK approach to modelling in many scenarios as long as the correct conclusions and limitations are included in the discussion. Clarity should be given that this model does not *prove* the full mechanism, it simply shows that *such a mechanism is capable* (or incapable) *of fitting the experimental data*. That does not conflict with the conclusion of kinetic modelling in this report: being that a diffusion-controlled coupling step is necessary to fit the $P(\text{tBu})_3$ data. That result still stands. As it is now though, the SI may give less familiar readers the impression that kinetics of the entire catalytic pathway have been delineated, which is incorrect.
13. In this reviewer's opinion, the authors have missed an (small, but still interesting) opportunity here to use results of the kinetic modelling to further their mechanistic understanding of the different catalytic systems. Best modelling practices usually dictate searching for the *minimum set of necessary equations* to fully describe an experimental system. The composition of this minimum set of equations should agree with and thus reinforce the kinetic orders observed through difference excess experiments. The ability to exclude, or requirement to include, certain kinetic steps in different models in order to achieve goodness of fit indicates the relevancy of those kinetic steps to the overall mechanism.
14. **$P(\text{tBu})_3$ System (Fig.4)**. This data can be fit with the minimal model below ("Fig4 PtBu3.cps"). This is in agreement with the proposed positive order in **1** in fig S7, but admittedly does not account for the negative order in **2**. This inhibition by **2** is clearly real, as demonstrated in figs S7 and S8, but it is interesting that this

simple model can still fit this set of data. I am looking forward to the authors next report on the “complex interplay between catalyst initiation and decay.”

15. **PEPPSI System (Fig. 7).** From the kinetic orders proposed in Fig S9 this system should be able to be described by the same simple model as above (Fig7 PEPPSI.cps), but even with the inclusion of diffusion-controlled coupling we can see that this is not the case. Including simple 1st order catalyst deactivation also didn't improve this fit.

With addition of equilibrium binding steps the model fits much better (Fig7 PEPPSI v2.cps):

The necessity to treat catalyst binding as an equilibrium step is an interesting result because it indicates that irreversible oxidative addition cannot be the only kinetically relevant step. Yet VTN analysis of the same excess experiments clearly gives an order of 1 for substrate **1**. If reductive elimination was also contributing to some rate control I would expect the observed order of **1** to drop below unity. This is an interesting result that I cannot fully explain and think I am trying to dig too much out of the limited data available. I will leave it to say that I am looking forward to read future publications from the authors detailing the complexities of these catalyst systems.

16. **RuPhos System (Fig. 6)**. This fit *sort of ok* with the equilibrium binding model from above (Fig6 RuPhos.cps), but clearly some behavior is missing. In this case it actually looks like product formation is occurring faster than the model would predict, similar to what was seen with P(tBu)₃ (but to a lesser extent here).

Indeed, if diffusion-controlled coupling is added to this model the fit gets noticeably better (Fig6 RuPhos v2.cps).

Could it be that this system also experiences diffusion-controlled coupling to some degree? I am aware that the authors model without diffusion-controlled coupling seems to fit this data better than mine, but I am uncertain which features in their model cannot be simplified to the above system. I am also slightly skeptical on how the size of data points in plot fig. S14a affects visual perception of the fit. It can be seen that the fit does improve between figs. S14a and S14b, especially for the product trend before 60 minutes.

From a qualitative look at this data we can see that product **6** is formed immediately at the start of the reaction with a remarkably constant rate, even when there is very little intermediate **4a** present. While not as immediately diagnostic as the P(tBu)₃ system, I challenge the authors that there may be some diffusion-controlled coupling here as well.

17. **XantPhos System (Fig. 5).** Why was this system excluded from the modelling approach!? When I first saw fig. 5 I was blown away by the ability to delineate all of these intermediate species and I was looking forward to seeing a model created for this. To sate my disappointment, I created a model (Fig5 XantPhos.cps), and it is entirely as satisfying as expected! Of course, the model is highly underdetermined.

But, I think there is some interesting interrogation of this model possible. We can force *supposedly* similar modes of Pd binding to have equal kinetics and see that the fit is still maintained (Fig5 XantPhos v2). And in

this case the fit looks fairly well-determined, with relatively lower std. deviation values for all of the kinetic constants compared to the previous model. Though, with reductive elimination being the rate controlling step for this chemistry it is not too surprising that the fit is not so sensitive to the relative binding kinetics.

Set equal kinetics: rxn1=rxn4=rxn7; rxn3=rxn8=rxn11

From this model we can look at the estimated rate constants of reductive elimination to get an interesting overview of the relative reactivities. I am not certain how useful this information is, or how robust it is considering it comes from a single set of experimental conditions, but I think it is interesting what can be extracted from this kinetic data if modelling is fully leveraged.

Step #	RE Position	Estimated Rate Constant
#2		1.38
#5		0.43
#6		0.33
#9		0.08

#10		0.64
#12		0.11

Comments on General Conclusions

18. This report is a very powerful demonstration of the understanding that can be gained through detailed reaction monitoring and kinetic analysis. From a technical analysis point of view the most impressive result is the ability to distinguish between the two nearly kinetically identical pathways of ring-walking and diffusion-controlled coupling for substrate **1**. A portion of this success is due to the correct choice of substrate: that which bears reactive sites accessible via solvation-sphere diffusion *but not* through ring-walking. Thus, it should be noted that technically diffusion-controlled coupling cannot be excluded as a pathway for substrates **7**, **10**, and **13** based on the current data.

I recommend the authors to include slightly more discussion on how the unique substrate (**1**) enables this analysis, to make it clearer for all readers who are not as well practiced in thinking through kinetic problems. This does not take anything away from the accomplishments of the kinetic method, but serves as a teaching opportunity to show how careful substrate choice is also a contributing factor to a study's success (and can also help readers identify opportunities for this type of kinetic differentiation in other studies).

Consider changing the penultimate conclusion sentence to something along the lines of "Furthermore, *careful substrate choice and the ability to conduct a mechanistic study in such a complex setting enabled the differentiation between ring walking behavior and diffusion-controlled coupling.*"

19. How did you arrive at the conditions used in fig. 11 to generate the maximum yield of the asymmetric **4a**? These initial conditions are entirely different than those used in the kinetic study, including a 20° difference in temperature. Did you use a kinetic model to help arrive at these conditions? Was it screening, guided by mechanistic understanding? Simply stating that "Reducing the amount of **2** and LiHMDS enabled us to access..." is not enough to fully explain how these new conditions were arrived at. This discussion should be expanded to help readers better understand your optimization process.

20. Substrate **13** has been demonstrated to inhibit ring-walking on its Xanthene core. The Pd(OAc)₂ / P(tBu)₃ catalyst system is the only one with proven diffusion-controlled coupling in addition to ring-walking. If Pd(OAc)₂ / P(tBu)₃ were applied to **13**, then I wonder if similar behavior as Fig. 10a would be observed, but with less **16a** (or none at all)? If so, then this would then be another very nice discrimination between ring-walking and diffusion-controlled coupling.

21. Building on these ideas, would it be possible to develop something like a "diffusion-control tag" consisting of a Pd-reactive moiety which could be appended to a substrate in a configuration inaccessible by ring-walking, to then enable discrimination between ring-walking and diffusion-controlled coupling? (An idea akin to the use of a radical clock to determine rates of radical reactions.) This may be a naïve approach, but even something as simple as attaching a bromobenzyl moiety to one of the open positions on the fluorene core of substrates **7**, **10**, or **13**. Or maybe something subject to beta-hydride elimination. It would still require careful kinetic analysis to interpret, but could be an interesting approach.

Below we have included a point by point response to the reviewers comments. Thank you again for the opportunity to submit this review manuscript.

Reviewer 1

The submitted manuscript by Berlinguette and Hein describes a series of studies on Hartwig-Buchwald Amination reactions to synthesize SpiroOMeTAD derivatives. While the work is interesting, it is too preliminary for publication currently. Moreover, given the narrow substrate scope and ligands investigated, even when fully fleshed out, I think this work is more appropriate for a specialized journal.

I have several significant concerns with the kinetic data and its interpretation:

- 1) Only one set of data was duplicated for each catalyst and ALL remaining experiments (most of the SI and all the conclusions in the paper) are based off a single set of data. This is not good practice and should not go into the literature this way. Every data should be (at least) duplicated.

Prior to publication we have indeed run multiple replicates of our data sets, validating both the standard error of our analytical hardware and the standard variation associated with our specific experimental protocol (atmosphere control, reagent order of addition and reagent

handling/purification). In the interest of brevity we did not include the entire volume of these control studies in our initial submission. However, our exceptionally high degree of reproducibility is evident on examining our variational experiments confirming the 0th order dependence for both the RuPhos and XantPhos. For examples, refer to revised SI - figures S12, and S13, which depicts four different experiments where initial concentrations of aryl bromide, amine and base are all being varied relative to standard conditions. The reaction time course profiles are in near perfect agreement until the reaction exhausts whatever material is held as limiting. We feel these experimental results echo the reproducibility and accuracy necessary to validate our conclusions. These results represent the mandatory threshold for good kinetic practice necessary to draw our conclusions.

We further contend that our results and conclusions are also in agreement with those observed by Buchwald and coworkers in a recent publication (see: *J. Am. Chem. Soc.* **138**, 12486-12493 (2016)). Thus, our results are supportive of relative order behaviour based on ligand effects, which has already been established for less complex systems.

- 2) The authors have not characterized ANY of the resting states and it's not clear why. These catalysts should be amenable to ³¹P NMR spectroscopic analysis with the phosphine-based ligands. Instead, the authors make assumptions IN EVERY CASE about what the resting state "must" be.

We acknowledge that the use of phosphine NMR has been widely used to determine catalyst resting states in palladium catalyzed cross coupling, however this comes with significant caveats that make its application in our reaction system somewhat inappropriate. The most significant is the presence of multiple competing catalytic processes. As our time course analysis displays, multiple Ar-Br species are in flux at any given time. NMR interpretation of the mixture of discrete complexes in this competitive mixture is highly open to interpretation and not conclusive. In addition, one of our comparative ligand systems (PEPPSI) lacks the necessary ³¹P. By contrast, our in-situ HPLC technique is the appropriate analytical strategy, as we sought to devise a comprehensive analysis of the operational reaction mechanism on a system displaying multiple competitive processes, and with a varied ligand subset.

Extracting power order rate laws and inferring catalyst resting state based on reaction progress kinetic data is well established and a fully vetted physical organic protocol, obviating single point isolation and analysis of catalytic intermediates (for reviews on this application see: *Angew. Chem. Int. Ed.* **44**, 4302-4320 (2005); *Angew. Chem. Int. Ed.* **55**, 2028-2031 (2016); *Angew. Chem. Int. Ed.* **55**, 16084-16087 (2016); *Chem. Sci.* **10**, 348-353 (2019); *J. Am. Chem. Soc.* **137**, 10852-10866 (2015)).

- 3) The authors assume reaction orders based (literally) on two data points. They have the "standard conditions" and one other concentration (either higher or lower). This is also not good practice. Of course, when you fit two points, you will always get a line (zero order, first order). But you could end up mis-assigning these orders or missing important information (e.g., saturation behavior, 2nd order, inhibition).

In a similar vein as our response to comment #2, we believe the reviewer has misinterpreted our analysis and is applying criteria applicable to traditional initial rate measurements. The kinetic fit and order determination is gleaned from the entirety of the time course data set of a reaction (typically 15-30 data points per reaction). This approach underpins the phenomenological data treatment central to RPKA and VTNA analysis - which are the central topic of the review papers cited in response #2.

The reviewer accurately points out that the order dependence we are reporting may not represent the catalytic behaviour under ALL possible sets of initial concentrations of ligand, substrate, base, nucleophile, etc. Rather, we are reporting the operational kinetic sensitivity for synthetically relevant reaction conditions typically employed for the synthesis of target chosen triarylamine products. The observed reaction order can and necessarily must vary as the relative ratio of reagent concentrations change (see *Top. Catal.*, **60**, 631 - 633 (2017)). This feature is the strongest argument against employing classical initial rate measurements over wide swings in initial substrate concentration. Again, we are not attempting to map the entirety of reagent order dependence over all possible combinations of initial concentrations, but rather validate the operational behaviour for synthetically relevant systems.

This notwithstanding, it is important to once again draw the reviewers attention to the reaction profiles in the revised SI - figures S10, S12, and S13. The order dependence we are testing and concluding is not the result of comparing only two experiments but the aggregate of all four. For example, let us focus on partially interpreting the reaction profiles seen in the revised supporting information for RuPhos (Figure S12). In this sequence, four independent experiments are carried out, yet no matter what species is changed (Ar-Br, LiHMDS, amine) the concentration vs time profiles denoting the consumption of Ar-Br (green - top left) and formation of product (blue - lower middle) are all identical between time = 0 and time = 60 min. Beyond this time point, catalysis ceases to follow this kinetic behaviour due to variation in stoichiometry and limiting reagent. However, given that four independent experiments all display identical rates in consumption of Ar-Br and formation of product, despite wide variation in initial conditions, the most accurate conclusion is that the catalytic system displays an overall zero order dependence on initial Ar-Br, LiHMDS, amine - so long as the initial concentrations vary in the ranges tested ($[Ar-Br] = 40 - 20 \text{ mM}$, $[LiHMDS] = 200 - 100 \text{ mM}$, $[amine] = 180 - 90 \text{ mM}$) for a given catalyst, ligand, solvent and temperature.

- 4) I encourage the authors to carefully read the Hartwig/Buchwald paper in JACS in 2006 (pg 3584) to understand more the complexities that are likely in these rate studies and why careful, controlled studies are important, in addition to having a clear understanding of the actual resting states.

The complexities of the amination study conducted in the paper cited had largely arisen due to catalyst activation/degradation behavior. This is why in our case, we opted to use Buchwald precatalysts (which do not suffer such poor activation behavior) as well as PEPPSI. As a result, the kinetic data obtained from our studies do not suggest the presence of these underlying complexities. We encourage the reviewer to be more specific in this context with respect to how we may have misinterpreted our data.

Finally, we do believe that complexities with respect to catalyst activation/degradation are present in the case of Pd(OAc)₂/P(tBu)₃ which led us to be more modest with our conclusions in the text of the paper. However, we felt it important to report these results without the use of a Buchwald precatalyst given the widespread use of this catalytic system in the synthesis of SpiroOMeTAD and triaryl amines more broadly.

- 5) Looking at the SI ... it looks like many of these reactions do not go to full conversion of 1. What is going on in those cases? Like it levels off at 50% or less.

The reactions are not able to go to full conversion due to varied concentrations of starting materials in order to probe reagent orders. Thus, stoichiometry does not allow for reactions to reach completion and is in fact an important control element verifying that our analytical tools are accurate and precise.

- 6) Table 1 really highlights my misgivings. The entire paper is premised on a clear understanding of the kinetic data but the rxn orders are only based on two points (I'm not even clear if the authors fit these decay profiles or just eye-balled it), and the turnover limiting steps are potentially wrong b/c they haven't identified the catalyst resting states. Basically, I'm not confident in the first four columns of data. The only thing I think they might have done correctly is assign whether it's ring-walking or not.

The method of analysis and our reasoning for its appropriate application has been detailed in our responses to comments #2, #3, #4. While it is regrettable that the reviewer is not versed in our technique, nor its intent, we acknowledge that misunderstanding (as evident from comments pertaining to how the reactions failed to reach conversion yet missed the absence of sufficient reagent to do so) point out that our message, methodology and core purpose of our study were not clear enough.

To address this shortcoming we have sought to clarify our central motivation; using our advanced analytical capabilities to delineate mechanistic complexity in the Buchwald Hartwig amination, which would normally be rendered irrevocable using classical techniques. These clarifying details now make up a new section in the introduction.

While we have executed a detailed kinetic analysis to categorize each ligand system with respect to its operational resting state, the observed power law orders are not pivotal to our observations or conclusion. Rather, our analysis and careful choice of substrate has allowed us to quantitatively validate the presence or absence of a critical mechanistic feature obfuscated to classical methods of analysis. To this end we thank the reviewer, as they agree with our assertion of which systems display ring walking.

- 7) Overall, each reaction uses superstoichiometric amounts of the amine coupling partner. Several studies (Larrosa et. al. Org Lett 2011, 13, 146 cited by these authors; Groombridge et. al. Chem. Comm. 2015, 51, 3832), use substoichiometric quantities of coupling partner to reduce false positives for ring-walking hits. Can the authors observe similar reactivity trends when using substoichiometric amine?

We have already carried out the experiment suggested by the reviewer. The set of different excess experiments (shown in Supporting info figures S8, S10, S12 and S13) reduce the [amine]₀ to a substoichiometric concentration relative to the aryl bromide (recalling that 4 equivalents are required per mol of aryl bromide). The results of these experiments show no change in intermediate distribution or reaction profile. Starving the system for amine nucleophile does not lead to a switch in which intermediate is formed, or even the rate of reaction in the case of RuPhos and PEPPSI. In all examples, the catalysis abruptly terminates when the amine is exhausted without the appearance of any previously unseen intermediates. RuPhos, PtBu₃, and PEPPSI never show the appearance of monocoupled 3, dicoupled 4a (opposite regioisomer), or tricoupled 5 - even at high conversion, near the timepoints when the reaction is near complete and amine is nearly fully exhausted. This is valid to the detection limit of our analytical technology, which can visualize components concentrations of 0.1mM with statistical significance. This invariant chemoselectivity demonstrates that ring-walking still occurs in the P(tBu)₃ (Supplementary Figure 8), PEPPSI-*IPr* (Supplementary Figure 10), and RuPhos (Supplementary Figure 12) systems with substoichiometric amounts of amine.

The reviewer is correct in as much as reducing the stoichiometry of the coupling agent can reduce false positives in the identification of ring walking catalyst, however, we also note that work by McNeil, et al. (see: ACS Macro Lett. 2016, 5, 69) has shown that these types of experiments can be misleading and lead to false positives when the intermediate displays greater reactivity than the starting material. In fact, time course data can help circumvent these issues.

To invalidate the possibility that a transient intermediate may have a much higher relative rate of cross coupling we have added a new study and section to our discussion. These data are presented in supplemental figure S20 (PEPPSI) and figure S## (RuPhos), where we directly measure the rates of reaction for the simplified dibromo substrate 7, and monobrominated product 8. Both substrates generated the same final product, 9, at nearly identical rates. This result confirms that the purported mono-coupled intermediates do not display aberrantly high rates of coupling, which could provide an alternative explanation for the lack of observed intermediate. Thus we are confident that our observation of ring walking is accurate, and that our method of analysis can provide an analytical validation for this mechanistic feature.

- 8) The authors comment on the difference between mono/bi dentate phosphines but only briefly mention differences between alkyl/aryl phosphines. It might be interesting to see the difference between mono/bi dentate with alkyl/aryl phosphines. For example, P(Ph)₃ vs P(tBu)₃ or, for Buchwald ligands DavePhos or Jackie Phos vs RuPhos.

We chose several different ligand classes to cover those commonly employed in the synthesis of triaryl amines to allow conclusions to be drawn and create a framework for where further study and classification could be carried out. Undoubtedly, differences in reactivity will arise as structural variations are implemented within these ligand classes. However, this lies outside the scope of the current paper as we needfully must draw a reasonable boundary on the current study. With the methodology and target system validated, deeper exploration into broad substrate variability, solvent, temperature, base strength and more can be now entertained. As there remains robust discussion (seen in our responses to the previous comments) it behooves all experimentalists to first agree that the physical organic practices are in place prior to such a wide survey.

- 9) The authors provide evidence for the fact that mono- vs bi-dentate ligands affect ring-walking, but don't rationalize these results. Can the authors provide some insight for why the mono-dentate ligands don't dissociate? (sterics seems like the easy answer) And why the NHC ligand is more comparable to the mono-dentate phosphines?

Based on the CTP literature, the ability to ring walk is a complex mixture of matching steric and electronic properties of the substrate and catalyst. We chose to avoid an in depth discussion of what may be governing the behavior in our system given the diversity of ligands chosen. This task would be best undertaken when undergoing small structural variation on the catalyst/substrate to probe its effect on the behavior. As such, we feel it would be overreaching to claim or attempt to rationalize any correlation between the denticity of the ligand and its propensity to ring walk.

- 10) Substrate scope – I understand that they chose the scope of SpiroTADs based on existing hole-transporting materials. However, I think more relevant/comparable substrates could have been selected to prove the generalizability of the ring-walking findings. For example, instead of using a ketone-substituted spirocycle (Csp2) for an “electron deficient” substrate, they could have chosen a di-fluorinated spirocycle (Csp3), which is used as a monomer for conjugated polymers. The authors also do not report an asymmetrically substituted derivative of substrate 13, which was pointed out in their introduction. Also, the authors cite the fact that the electron-rich amine should theoretically deactivate the spirocyclic core to OA as additional evidence for ring-walking. Given that the amine can “inhibit” reaction rate for coupling performed with P(tBu)₃ I'm curious if/how the electronics of the amine affect the rate determining step and/or ring-walking

We do not believe that expanding the scope of substrates explored in this publication will significantly increase its impact as its focus was largely focused on mechanistic information rather than method development. However, comments like this demonstrate that our study is compelling and worth expanding upon!

The discussion of the desymmetrized substrate 13 in the introduction was meant to serve as an example of how incorporating different functional groups in a hole transport material and reducing its symmetry could improve its electronic and physical properties. We chose to focus our efforts on the tetrabrominated spiro core as we believed it to be an inherently more difficult substrate. The latter starting material has 4 Ar-Br which are identical to one another. The ability to differentiate these and gain selectivity in such a system is an exceptionally high bar and enables entry into chemical space which is otherwise exceedingly difficult to access. In contrast, 13 is an electronically biased system with the xanthene core being more electron rich relative to the fluorene, thus allowing an avenue for differentiating the different Ar-Br bonds. Furthermore, access to derivatives of 13 containing different halogens on both ring systems are readily accessible due to their straightforward synthesis.

- 11) Probing diffusion vs ring-walking: The authors are missing some examples from the conjugated polymer literature using capping agents and/or catalyst traps to differentiate between diffusion based coupling and ring walking by analyzing polymer end-groups for catalyst-transfer

polymerization. For example, please see: Koeckelberghs and coworkers Polym. Chem. 2011, 49, 5339. McNeil and coworkers JACS 2018, 140, 7846.

These studies do not provide experimental evidence that distinguishes between authentic ring walking behavior versus diffusion-controlled coupling where the catalyst remains within the solvation sphere of the electrophile. The implementation of 'catalyst trap' provides circumstantial evidence for ring walking where the lack of reactivity of the catalyst with a more activated electrophile (such as an aryl iodide) provides evidence that the catalyst must remain bound to the growing pi system. However, the same would be true if the catalyst simply remains within the solvation sphere of the growing pi system and thus this methodology cannot distinguish between these two mechanistically distinct pathways.

12) This acronym is never spelled out: SpiroOMeTAD

We have updated the introduction and given the full IUPAC name of SpiroOMeTAD in the introduction of the manuscript: "A small library of asymmetric 2,2',7,7'-tetrakis[N,N-di(4-methoxyphenyl)amino]-9,9'spirobifluorene (SpiroOMeTAD) derivatives were successfully synthesized..." (page 1).

13) In Figure 2A, you have what should be a Pd(III) intermediate (after deprotonation) or its anionic Pd(II) or maybe it doesn't exist? See Hartwig/Buchwald paper mentioned above.

We have fixed this error and updated Figure 2A in the manuscript.

Reviewer 2

Recommendation: Accept with minor revisions

This is a solid and well-designed kinetic study into the effect of catalyst speciation on Pd ring-walking in BHA reactions. The authors have demonstrated the unique ability to distinguish between two pathways which are nearly kinetically identical. In doing so they have revealed interesting mechanistic insight into the relationship between catalyst/ligand structure and substrate interaction. In a critical step which is often missing from mechanistic investigations, they have then used this knowledge to improve upon the original process and enable selective access to previously inaccessible asymmetric products. The substrates also have their own significance to the discipline of materials science. This report will thus be of interest to scientists from a range of disciplines and is strongly worthy of publication.

The authors primary conclusions are supported with well-designed experimental evidence and chemical reasoning, with which I am in agreement. No further experimental work is needed. The following points should be considered such that the manuscript might meet a high level of scrutiny. None of these points will change the broader conclusions but will serve to help clarify and demonstrate the principles of rigorous kinetic investigation to any future readers. I offer these comments in good faith and welcome dialogue with the authors on any of these points if I have misinterpreted their results.

I strongly recommend that the authors address points # 5, 6, 7, 10, 11, 12, 18, and 19 before this manuscript is published. The remaining points are left to author and editor discretion.

We thank the reviewer for their exhaustive and thorough review of our work. We are encouraged by their comments, their insight and their appreciation of our method. We have incorporated a variety of their comments, which, we believe, has strengthened the article and our communication.

General Editorial Comments

1. In Fig 1a, at first glance I confused the right-pointing arrows with retrosynthesis arrows. Upon closer inspection the intended meaning was clear, but this may confuse other readers too.

We have opted not to change the arrow structure, but would like input if the intended meaning is still unclear.

2. The first paragraph in section “Ligand effects on SpiroOMeTAD synthesis” is a giant wall of text and is not the most pleasant to read. Can it be broken into two smaller paragraphs?

We have redrafted the introduction to better communicate the experimental intent and our rationalization. This is now composed of a description of the intended analytical outcomes and how they would tie to mechanistic control elements.

3. In Fig.10a I think some of the “R” group assignments in the legend need to be fixed. Currently it appears that **14a** and **14b** are the same, and that **16a** and **16b** are the same.

We have fixed this error and updated Figure 10 in the manuscript.

4. Fig. S3 is missing description of the initial conditions, which are present in Figs. S4, S5, & S6 for the other pre-catalysts.

We have added a description of the initial conditions to Supplementary Figure 3 in the SI.

5. I strongly suggest including a supplementary data file including all of the raw concentration vs time data that was used to construct the figures shown in the manuscript and SI. Sharing this data demonstrates good faith in transparency for kinetic analysis and makes the job of any reviewers or future interested parties much easier.

We have included the concentration versus time data in Excel sheets for each ligand and aryl bromide system. We have also included the COPASI files.

Comments on Kinetic Analysis

6. What is the basis for the proposed positive order in LiHMDS for the P(tBu)₃ system? In figure S7, the “Increased LiHMDS” reaction can be seen to consume **1** faster, but formation of **4a** and **6** are actually both *slower* than under standard conditions! This suggests that LiHMDS does not have a

positive order on the product forming reaction, but is instead causing degradation of **1**. In fact, at the end of this reaction there is only *ca.* 17mM **6** and 12mM **4a**, leaving a missing balance of 9mM, or ~24% of the initial loading of **1**. It is dangerous to make conclusions on kinetics experiments with only 76% of mass balance accounted for.

At the end of the data for this reaction the conversion of **4a** to **6** has also nearly stopped. Is this because the increased LiHMDS is also decomposing **2**, or is it affecting catalyst activity?

We apologize for the error, results from an increased LiHMDS experiment were erroneously mixed with results from a decreased LiHMDS experiment in Supplementary Figure 8 (previously Supplementary Figure 7). We have corrected this figure to include only the data from a reduced LiHMDS experiment (with $[\text{LiHMDS}]_0 = 120 \text{ mM}$). These trends show significantly decreased rates of starting material consumption, intermediate formation, and product formation, corroborating a positive order in LiHMDS. We have also added both standard conditions timecourses to this figure to demonstrate that even with the variance in the $\text{P}(\text{tBu})_3$ conditions, there is clearly a positive order in LiHMDS.

7. In Fig. S7, what is going on with mass balance for the “Reduced **2**” experiment? At the last data point there is *ca.* 28mM **6** and 12.5mM **4a**. Should this then require $(2 \times 12.5) + (4 \times 28) = 137\text{mM}$ of **2** having been present at the start? But the legend for this figure says that experiment only got $[\mathbf{2}]_0 = 90\text{mM}$?

Again, this error stems from a mistake in data labeling, not in the mass balance. The experiment in question was performed with $[\text{amine}]_0 = 120 \text{ mM}$ not the labeled amount of 90mM. The final mass balance for this experiment was $[\mathbf{6}]$ *ca.* 28 mM and $[\mathbf{4a}]$ *ca.* 9 mM (please refer to attached excel files for full reference). Thus, the mass balance relative to the input **1** (aryl bromide) is ~38 mM analytically accounted in product with ~4mM remaining as **1** (*cf.* the intended 40mM input). This variation is the result of our mass-balance centered approach to estimate the approximate extinction coefficients for multiple samples from parallel kinetic experiments (see SI section “Conversion of peak area to concentration”). Our method utilizes linear algebra and reduced mean error variation to approximate component response factor. This method was adopted to mitigate spectroscopic and instrumental variation inherent to operational reality of experiments that spanned months of research. This technique returns values within 5% deviation, which is well below the threshold needed to confidently support the conclusions in this paper.

This variation is also the reason that a deeper dive into kinetic modeling is - for the time being - not a central focus of our report (*vide infra*). These facts notwithstanding, we submit that the time course observations, rate profiles and patterns of reactivity which we have elucidated represent the core of our novel approach. That fingerprint kinetic analysis can be used to delineate mechanistic details in the absences of species involvement in rate determining catalytic steps.

In the ensuing months we have begun adapting our method now to include a standard addition protocol utilizing dosing of authentically isolated compounds to augment and reinforce our linear algebra method. These results will be the centerpiece of dedicated study to a specialized journal. For this submission, the figure error has been corrected in Supplementary Figure 8 (previously Supplementary Figure 7).

8. In Figs. S9, S10, and S11, in the plots of **[1]** vs time, for the $[1]_0=20\text{mM}$ reaction, the trends of $[1]$ itself have been concentration-adjusted up by 20mM so that they overlay with the initial condition of the other three experiments. I believe this was done to clearly demonstrate the comparison of conversion rate between these conditions. This is OK, but there needs to be a very clear note in the figure caption describing this manipulation and that these triangle data points are not the true values of $[1]_t$. Otherwise it can lead to confusion for someone less well versed in this type of analysis, who may interpret these plots to indicate that the reaction stalls with 20mM **1** remaining.

An additional figure (Supplementary Figure 7) and an explanation were added to the SI to clarify this point (page S10). Additionally, the trends in the relevant SI figures were relabeled 'Reduced 1 (Adjusted)' and a sentence was added to each figure caption reading, "see Supplementary Figure 7 for an explanation of why the 'Reduced 1' trend in the $[1]$ versus time plot starts at 40 mM".

9. In Fig S9 (PEPPSI) the caption indicates that the reduced **2** experiment also had reduced LiHMDS (90mM and 100mM, respectively). However, this is not the case for the reduced **2** experiments for any of the other catalyst systems. Is this a typo in one or more of the figure captions, or is it how the experiments were actually performed? If it is real, then why was this change made for PEPPSI but not the other catalysts? I suspect it is because "significant inhibition was observed when LiHMDS was used in large excess relative to **2**." If so, please see my next point.

This figure was accidentally omitted from the SI. The SI now includes Supplementary Figure 11, which shows the timecourse data of the experiment with $[\text{amine}]_0 = 90 \text{ mM}$ and $[\text{LiHMDS}]_0 = 200 \text{ mM}$. There is barely any conversion after six hours under these conditions, indicating catalyst death. When the reaction is run with reduced base and reduced amine, a clear 0th order dependence in amine is observed.

10. In the manuscript it is mentioned in two places that the PEPPSI system experiences inhibition by LiHMDS, with a reference made to the SI. However, I was not able to find the data supporting this statement in the SI. Fig S9 depicts zero-order kinetics in LiHMDS, and I cannot find any kinetic results elsewhere that describe the behavior if LiHMDS is used in significant excess in the PEPPSI system. Perhaps this was accidentally left out of the SI?

As in response 9, this was an omission and now rectified.

Comments on COPASI Modelling

11. The parameters used to fit the two models to the plots in fig S14 should be reported in the SI in addition to the structure of the model. Standard practice for any scientific report is to include the necessary information for the experiments to be repeated & verified by a third party. If a kinetic model is used in an investigation, *especially* if one of the major conclusions rests on results of that model, then the full details of that model must be made available. At a minimum this includes 1) the structure of the model, 2) the model parameters, and 3) fitting results of that model to relevant experimental data. Ideally this should also include some limited demonstration of goodness-of-fit / confidence statistics (or similar).

We have updated our reporting of the kinetic model, and the logic associated with conclusions drawn from various extensions of a new minimal basis set of reactions. These minimal models are based on the reviewer's comments, contextualized by the new discussion section we have added. The intent here was to walk through the various complexities that must be added to our proposed system in order to recapitulate the data. We are exceptionally grateful to the reviewer for the conversation and for challenging us to dig deeper into the modeling interpretation. This exercise has not only clarified our study's findings, but serves to further reinforce the accuracy of our analytical trends; it is noteworthy that such simple kinetic models can produce such accurate model fits. The excellent agreement between experiment and simple kinetic model would be very difficult to achieve if systematic errors were present in our experimental execution and data reduction.

12. The COPASI model used to fit this data is much more complex than it needs to be (vide infra). Based on the kinetic conclusions in Figs. S7, S9, S10, and S11, the majority of steps depicted in scheme S1 are not kinetically relevant and estimates of their kinetic constants from the standard reaction data will not be meaningful. I suspect that the model as applied to the data in fig. S14 is highly underdetermined with significant errors/uncertainties $\gg 100\%$ associated with most of the parameter estimations. This is still an OK approach to modeling in many scenarios as long as the correct conclusions and limitations are included in the discussion. Clarity should be given that this model does not *prove* the full mechanism, it simply shows that *such a mechanism is capable* (or incapable) *of fitting the experimental data*. That does not conflict with the conclusion of kinetic modeling in this report: being that a diffusion-controlled coupling step is necessary to fit the $P(tBu)_3$ data. That result still stands. As it is now though, the SI may give less familiar readers the impression that kinetics of the entire catalytic pathway have been delineated, which is incorrect.

As stated in our answer to point 12 - we have fully rebuilt this part of the manuscript to address this comment.

13. In this reviewer's opinion, the authors have missed an (small, but still interesting) opportunity here to use results of the kinetic modelling to further their mechanistic understanding of the different catalytic systems. Best modelling practices usually dictate searching for the *minimum set of necessary equations* to fully describe an experimental system. The composition of this minimum set of equations should agree with and thus reinforce the kinetic orders observed through difference excess experiments. The ability to exclude, or requirement to include, certain kinetic steps in different models in order to achieve goodness of fit indicates the relevancy of those kinetic steps to the overall mechanism.

As comments #13 - 17 deal with deeper interpretations of the model, we have prepared a broader commentary, explaining our changes, and where we diverge from the reviewer's interpretation. Our responding commentary is presented below but addresses comments 13-17 as a unit.

We are deeply grateful to the reviewer for the time and thought that they have offering in the conversation on the application and interpretation of the model (highlighted in comments #13 - 17). While many relevant data extraction and extrapolations are possible, we remain hesitant to include the

full depth of this analysis in this current work. As this reviewer has highlighted, a real (or even perceived) fault of such analysis remains the possibility of advancing an overdetermined model, where fit values do not unambiguously eliminate certain mechanistic hypothesis, or where modeled rate constants disagree with the observed resting state and order observed through VTNA and time course analysis.

At this time, even given our best efforts, the complexity of the systems provide sufficient uncertainty that we are not yet convinced that the simplified models (as put forward by the reviewer) are appropriate to extract the level of detail we would be satisfied with. This rationalization is predicated by two main arguments

- 1) Our intention with this study was to solidly demonstrate to a broad audience the value complex reaction progress analytical approaches can play, when coupled with appropriate model systems, in the delineation of entangled catalytic processes. In this case, the underlying mechanistic features control the ease by which next generation conductive electronic materials can be accessed. This discussion touches on long held preconceptions as to the limitation both in our analytical capability, validity and method of experimentation (as highlighted by our conversion and response with Reviewer #1). To access and reach the broadest possible audience in this first report, we chose to focus on topical interpretations that were solidly grounded, and save the more detailed modeling work for a specialized study. Part of the complexity in the current work stems from our aim to provide a mechanistic survey across multiple substrates, reaction conditions and ligands; each with their own nuanced variation that does not easily transfer to a single communication. We echo the reviewers comments - we are excited by the deeper studies this work unlocks and look forward to their completion.
- 2) With the data we currently have, some conclusions are absolutely valid, while some remain open to interpretation. With our current data, we are confident that the modeling demonstrates several features. These include:
 - a) The ability of a simplified minimal model to accurately reproduce the observed chemoselectivity and intermediate profiles for the PEPPSI, RuPhos and PtBu₃ systems strongly supports our conclusion that ring walking across the conjugated pi system is at play. The only other conceivable rationale to this observation would be if the system “shortcuts” the mono- and tri-functionalized products due to an exceptionally high rate constant for CN coupling unique to these intermediates. Our new relative study (discussed in the new section “Alternative Mechanistic Interpretations” and in SI figures 20 and 21) solidly dismiss this possibility. Thus we are comfortable including this discussion and the introduction of the minimal models with the conversion leading readers through their application.
 - b) The necessary inclusion of a diffusion limited coupling event for the PtBu₃ system is also well supported. As our original model showed, without this feature the concentration profile for product and dicoupled intermediate would be impossible. The simple A-> B -> C stepwise model can not account for our observations and thus we are compelled to expand the model to include a

diffusion limited direct coupling to account for this. We have now included this model to the manuscript, incorporating the same discussion now substituting a modified minimal kinetic model. The conclusion and visualization remains unchanged from our original submission, as the extraneous reactions in our original model have simply been collapsed into the more reduced framework.

We can not yet conclusively add all of the points of discussion raised by the reviewer.

- c) While the reviewer's observation that the fit appears to improve upon inclusion of diffusion limited coupling in the RuPhos system, we do not believe this is fully justified as of yet. Attached below is one example where we have applied a parameter optimization using the Reviewer's model, with and without the diffusion step. These are the results by applying the evolutionary programming optimizer in COPASI, starting from randomly selected parameters and allowing both to converge to a similar best fit error, where the weighted values to the time dependent data was normalized to 1 for all three concentration profiles (in place of the autoselected weighting). In this case, there is not sufficient visible variation between the two implied models to definitively rank or exclude either.

# ^	Name	Reaction
1	SM Binding	$SM + Cat = SM_Cat$
2	Int Formation	$SM_Cat \rightarrow DiSame + Cat$
3	DiSame Binding	$DiSame + Cat = DiSame_Cat$
4	Product Formation	$DiSame_Cat \rightarrow Prod + Cat$
	New Reaction	

# ^	Name	Reaction
1	SM Binding	$SM + Cat = SM_Cat$
2	Int Formation	$SM_Cat \rightarrow DiSame + Cat$
3	DiSame Binding	$DiSame + Cat = DiSame_Cat$
4	Product Formation	$DiSame_Cat \rightarrow Prod + Cat$
5	Diffusion Control Product Formation	$SM_Cat \rightarrow Prod + Cat$
	New Reaction	

d) A deeper interpretation, including extracting unambiguous relative rates for the intermediate processes, such as that listed for the XantPhos system suggested by the reviewer, would be an overreach at this current time. While we are absolutely confident that the system is well behaved to the degree that the conclusions we have put forward are valid and justified, there remains sufficient uncertainty such that this further analysis may be open to interpretation.

Overall, we absolutely share the enthusiasm of the reviewer and are hard at work improving our data quantification, systemic control to provide meaningful modeling data.

14. **P(tBu)₃ System (Fig.4).** This data can be fit with the minimal model below (“Fig4 PtBu3.cps”). This is in agreement with the proposed positive order in **1** in fig S7, but admittedly does not account for the negative order in **2**. This inhibition by **2** is clearly real, as demonstrated in figs S7 and S8, but it is interesting that this simple model can still fit this set of data. I am looking forward to the authors next report on the “complex interplay between catalyst initiation and decay.”

15. **PEPPSI System (Fig. 7).** From the kinetic orders proposed in Fig S9 this system should be able to be described by the same simple model as above (Fig7 PEPPSI.cps), but even with the inclusion of diffusion-controlled coupling we can see that this is not the case. Including simple 1st order catalyst deactivation also didn’t improve this fit.

With addition of equilibrium binding steps the model fits much better (Fig7 PEPSI v2.cps):

The necessity to treat catalyst binding as an equilibrium step is an interesting result because it indicates that irreversible oxidative addition cannot be the only kinetically relevant step. Yet VTN analysis of the same excess experiments clearly gives an order of 1 for substrate **1**. If reductive elimination was also contributing to some rate control I would expect the observed order of **1** to drop below unity. This is an interesting result that I cannot fully explain and think I am trying to dig too much out of the limited data available. I will leave it to say that I am looking forward to read future publications from the authors detailing the complexities of these catalyst systems.

16. **RuPhos System (Fig. 6)**. This fit *sort of ok* with the equilibrium binding model from above (Fig6 RuPhos.cps), but clearly some behavior is missing. In this case it actually looks like product formation is occurring faster than the model would predict, similar to what was seen with $P(tBu)_3$ (but to a lesser extent here).

SM + Cat = SM_Cat
 SM_Cat -> DiSame + Cat
 DiSame + Cat = DiSame_Cat
 DiSame_Cat -> Prod + Cat

Indeed, if diffusion-controlled coupling is added to this model the fit gets noticeably better (Fig6 RuPhos v2.cps).

SM + Cat = SM_Cat

SM_Cat -> DiSame + Cat

DiSame + Cat = DiSame_Cat

DiSame_Cat -> Prod + Cat

SM_Cat -> Prod + Cat

Could it be that this system also experiences diffusion-controlled coupling to some degree? I am aware that the authors model without diffusion-controlled coupling seems to fit this data better than mine, but I am uncertain which features in their model cannot be simplified to the above system. I am also slightly skeptical on how the size of data points in plot fig. S14a affects visual perception of the fit. It can be seen that the fit does improve between figs. S14a and S14b, especially for the product trend before 60 minutes.

From a qualitative look at this data we can see that product **6** is formed immediately at the start of the reaction with a remarkably constant rate, even when there is very little intermediate **4a** present. While not as immediately diagnostic as the $P(tBu)_3$ system, I challenge the authors that there may be some diffusion-controlled coupling here as well.

17. **XantPhos System (Fig. 5).** Why was this system excluded from the modelling approach!? When I first saw fig. 5 I was blown away by the ability to delineate all of these intermediate species and I was looking forward to seeing a model created for this. To sate my disappointment, I created a model (Fig5 XantPhos.cps), and it is entirely as satisfying as expected! Of course, the model is highly underdetermined.

#01 SM Binding	$SM + Cat = SM_Cat$
#02 Mono Formation	$SM_Cat \rightarrow Mono + Cat$
#03 Mono Binding Same	$Mono + Cat = Mono_Cat_Same$
#04 Mono Binding Opp	$Mono + Cat = Mono_Cat_Opp$
#05 DiSame Formation	$Mono_Cat_Same \rightarrow DiSame + Cat$
#06 DiOpp Formation	$Mono_Cat_Opp \rightarrow DiOpp + Cat$
#07 DiSame Binding	$DiSame + Cat = DiSame_Cat$
#08 DiOpp Binding	$DiOpp + Cat = DiOpp_Cat$
#09 Tri Formation 1	$DiSame_Cat \rightarrow Tri + Cat$
#10 Tri Formation 2	$DiOpp_Cat \rightarrow Tri + Cat$
#11 Tri Binding	$Tri + Cat = Tri_Cat$
#12 Prod Formation	$Tri_Cat \rightarrow Prod + Cat$

But, I think there is some interesting interrogation of this model possible. We can force *supposedly* similar modes of Pd binding to have equal kinetics and see that the fit is still maintained (Fig5 XantPhos v2). And in this case the fit looks fairly well-determined, with relatively lower std. deviation values for all of the kinetic constants compared to the previous model. Though, with reductive elimination being the rate controlling step for this chemistry it is not too surprising that the fit is not so sensitive to the relative binding kinetics.

Set equal kinetics: rxn1=rxn4=rxn7; rxn3=rxn8=rxn11

From this model we can look at the estimated rate constants of reductive elimination to get an interesting overview of the relative reactivities. I am not certain how useful this information is, or how robust it is considering it comes from a single set of experimental conditions, but I think it is interesting what can be extracted from this kinetic data if modelling is fully leveraged.

Step #	RE Position	Estimated Rate Constant
#2		1.38
#5		0.43
#6		0.33

#9		0.08
#10		0.64
#12		0.11

Comments on General Conclusions

18. This report is a very powerful demonstration of the understanding that can be gained through detailed reaction monitoring and kinetic analysis. From a technical analysis point of view the most impressive result is the ability to distinguish between the two nearly kinetically identical pathways of ring-walking and diffusion-controlled coupling for substrate **1**. A portion of this success is due to the correct choice of substrate: that which bears reactive sites accessible via solvation-sphere diffusion *but not* through ring-walking. Thus, it should be noted that technically diffusion-controlled coupling cannot be excluded as a pathway for substrates **7**, **10**, and **13** based on the current data.

I recommend the authors to include slightly more discussion on how the unique substrate (**1**) enables this analysis, to make it clearer for all readers who are not as well practiced in thinking through kinetic problems. This does not take anything away from the accomplishments of the kinetic method, but serves as a teaching opportunity to show how careful substrate choice is also a contributing factor to a study's success (and can also help readers identify opportunities for this type of kinetic differentiation in other studies).

Consider changing the penultimate conclusion sentence to something along the lines of "Furthermore, *careful substrate choice and* the ability to conduct a mechanistic study in such a

complex setting enabled the differentiation between ring walking behavior and diffusion-controlled coupling.”

This comment is well taken and seems reflected in the comments received by reviewer 1 and 3. As a result, a significant addition to the text was made delineating the different scenarios which could be present and how to interpret them. Moreover, a small addition to conclusion was made to highlight the importance of judicious substrate choice, coupled with a robust monitoring technique.

19. How did you arrive at the conditions used in fig. 11 to generate the maximum yield of the asymmetric **4a**? These initial conditions are entirely different than those used in the kinetic study, including a 20° difference in temperature. Did you use a kinetic model to help arrive at these conditions? Was it screening, guided by mechanistic understanding? Simply stating that “Reducing the amount of **2** and LiHMDS enabled us to access...” is not enough to fully explain how these new conditions were arrived at. This discussion should be expanded to help readers better understand your optimization process.

We used the results of our mechanistic study to optimize conditions for the synthesis of intermediate **4a**. We chose the [amine]₀ to arrest reaction progress at the ideal [**4a**] = 20 mM and [**6**] = 5 mM. The remaining changes were made to allow more expedition use of our limited catalyst on hand - we reduced catalyst loading to 2.5 mol%, while decreasing the [LiHMDS]₀ accordingly to prevent deactivation of the PEPPSI-IPr precatalyst. While the final reaction conditions were derived from an aggregate of mechanistic insight (and we do not show the specific links from each experiment), the design of conditions were rational based on the extensive control and reaction optimization we had completed.

20. Substrate **13** has been demonstrated to inhibit ring-walking on its Xanthene core. The Pd(OAc)₂ / P(tBu)₃ catalyst system is the only one with proven diffusion-controlled coupling in addition to ring-walking. If Pd(OAc)₂ / P(tBu)₃ were applied to **13**, then I wonder if similar behavior as Fig. 10a would be observed, but with less **16a** (or none at all)? If so, then this would then be another very nice discrimination between ring-walking and diffusion-controlled coupling.

We absolutely agree that this substrate has enormous mechanistic potential. At this time to make sure such an analysis is properly scoped we endeavor to commit more resources, but at this time we are concerned that it would expand too far into a full separate story.

21. Building on these ideas, would it be possible to develop something like a “diffusion-control tag” consisting of a Pd-reactive moiety which could be appended to a substrate in a configuration inaccessible by ring-walking, to then enable discrimination between ring-walking and diffusion-controlled coupling? (An idea akin to the use of a radical clock to determine rates of radical reactions.) This may be a naïve approach, but even something as simple as a attaching a bromobenzyl moiety to one of the open positions on the fluorene core of substrates **7**, **10**, or

13. Or maybe something subject to beta-hydride elimination. It would still require careful kinetic analysis to interpret, but could be an interesting approach.

This is an interesting idea and certainly worth giving more thought. However, we feel this lies beyond the scope of this publication. We will pursue the development of a “diffusion-control tag” for future publications in this area but at this time we feel it is beyond the scope of this first report.

Reviewer 3

In this manuscript, the authors investigate the ability of a Pd catalyst to “walk” along the backbone of an aromatic moiety after a successful C-N coupling to carry out a second C-N coupling. This ring walk has been concluded and studied earlier and the authors now take a look at the particular case of tetrabrominated 9,9'-spirobifluorenes. The goals of this work are to provide further support of a ring walk, to investigate its ligand dependency, and to apply it in the synthesis of unsymmetric SpiroOMeTAD derivatives. The latter are of importance for functional materials research and hard to synthesize otherwise. The authors discuss that for a double C-N coupling at a 1,4-dibromoaryl unit (and analogous also at a 4,4'-dibromobiphenyl moiety) three pathways A–C can be followed: A) via the ring walk, B) via a solvation sphere that keeps the catalyst close, and C) under diffusion control. The latter refers to a complete diffusion and reapproach of the catalyst before the second (third, fourth) C-N coupling. The authors argue that the observation of only the intermediate at which both C-N couplings had occurred at the same biaryl (fluorene) moiety supports this ring walk scenario. Indeed, having the second C-N coupling occur at the same biaryl system would be in agreement with the catalyst not detaching from this unit. The second oxidative addition at the aminobiaryl unit would be otherwise electronically disfavored and slower than a second oxidative addition at the remaining, more electron-poor dibromobiaryl unit. However, the ring walk is only one possible explanation and there may be other causes for the observed preferences. The authors have recorded several sets of kinetic data with four different ligands that show varying behavior: P(tBu)₃ and RuPhos show only intermediate 4a (both C-N couplings at the same biaryl unit) whereas Xanthphos and a PEPPSI ligand show a mixture of all possible mono, bi, and trisubstituted intermediates. In addition, for P(tBu)₃ the formation of 4a is slower than the product formation, which indicates two product forming pathways. A COPASI simulation using a model involving all potential intermediates and an addition equilibrium bypassing intermediate 4a led to a good overlay of the curves. For the other ligands modeling this bypass was not necessary to achieve a good overlay. The catalyst system providing the highest amount of 4a as intermediate is also used to prepare the unsymmetric products 18–21, which underlines the synthetic usefulness for making unsymmetric SpiroOMeTAD units.

I agree that the new conditions for the synthesis of unsymmetric SpiroOMeTAD derivatives using the PEPPSI catalyst greatly facilitates the synthesis of such compounds. However, whether such unsymmetric compounds have improved redox/hole transport properties for applications in materials still remains to be shown.

We believe that testing the properties of the asymmetric derivatives synthesized in this study lies outside the scope of the current publication. The goal of this study was to leverage a mechanistic understanding of SpiroOMeTAD synthesis to enable rapid access to asymmetric derivatives. This work

will enable the materials chemistry community to easily probe this chemical space to optimize properties. We expect this to be a fruitful area of research given the recent reports delineated in the introduction of our paper highlighting the importance of asymmetric substitution patterns.

Regarding Hartwig-Buchwald couplings in general, the insight gained from this study is only of moderate interest, because the ring walk has been discussed earlier for such C-N couplings. Even for 4,4'-dibromofluorenes and a Pd-P(tBu)₃ catalyst, this ring walk has already been investigated (see Ref.: 26, DOI: 10.1039/c8py01646a). Therefore, I see this study as an expansion on earlier works with relevance to the particular substrate class of SpiroOMeTADs, but it is not of greater relevance.

We believe the reviewer has fundamentally misunderstood the impact of this work. Ring walking is an incredibly important phenomenon which underlies an entire field of polymerization. Moreover, as demonstrated in our study, provides an avenue for selectivity in small molecule synthesis as well. Despite this importance, the evidence for ring walking remains largely circumstantial (we point the reviewer to a recent review on the topic: Trends Chem., 2, 493-505 (2020)). Thus, the impact of our study should not be viewed as a simple extension of the work of Suranna et al. Instead, it should be viewed as the strongest evidence to date of the mechanistic phenomenon of ring walking. Furthermore, we highlight the importance of using a complex model system with 2 separate pi systems in teasing apart ring walking behavior vs diffusion controlled coupling providing a platform for future studies. Finally, we leveraged the mechanistic information gathered to achieve unprecedented selectivity, further demonstrating the importance of the mechanistic data gathered.

I further have my reservations regarding the kinetic experiments and conclusions drawn from them. The data may be in agreement with the proposed ring walk, but they may also have been overinterpreted. For example, the model used in the COPASI simulations is still quite complex. It contains numerous rate constants that would likely show linear dependencies if they were estimated using the limited acquired data. Moreover, the model used for the simulation of the Pd-P(tBu)₃ system is a complex variant of a simple two-path system, which could already be sufficient to simulate the observed curves. path 1: substrate  intermediate 4a  product; path 2: substrate  product. On the other hand, the formation of 6 involves four C-N couplings, each proceeding via multiple individual steps, which makes a correct simulation of the system highly challenging. The available data may therefore not be sufficient to draw reliable conclusions regarding the potential ring walk. The kinetic preference for the formation of 4a over other intermediates that is observed with certain ligands could be caused by other effects. It is not a direct proof of the ring walk. The Supporting Information is detailed, but it contains inconsistencies and seems to be a preliminary version (still parts highlighted in yellow). In addition, I have minor remarks (see below).

We have now revised our modeling section as detailed in our response to Reviewer #2. We do agree that the initial complete model provides added challenges to interpret. To address this case we have clarified both our application of the model, and our interpretation, giving rise to our three general categories summarized in the new figures 8, 9 and 10. These new minimal models do not suffer from the potential to over-determine the parameter estimation and agree uniformly over the various initial concentrations for each ligand that we have explored. More importantly, we have clarified the

conclusion drawn from these data - a) Ring walking remains the best explanation for the relative rate and intermediate product distribution and is the only assumption that must be added to a minimal kinetic model for PEPPSI and RuPhos; b) diffusion must be taken into account to recapitulate the product trends for PtBu₃; c) Xantphos operates in a regime akin to canonical, sequential CN coupling. We have also extended our discussion on how these trends extend to more complex Ar-Br systems as well as other nucleophiles - suggesting these are cross cutting trends. Finally, we have added details to highlight that both our method of analysis, experimental design and interpretation remains one of the first and only approaches to delineate a level of detail and complexity previously unseen and underappreciated in Pd-catalyzed CN cross coupling.

In addition, we have executed a dedicated mechanistic probe, comparing the relative rate of CN coupling on dibrominated substrate **7** vs. the potential mono-substituted intermediate **8** (Figure 11). This directly measures the rate of coupling for the both substrates using the PEPPSI and RuPhos catalysts and conclusively demonstrates that no aberrant rate acceleration exists, leaving ring-walking as the remaining rationale for the direct formation of **9** from **7** and absence of intermediate **8** in the PEPPSI and RuPhos systems.

We have further addressed labeling and figure errors - we are deeply apologetic for these oversights. The new supporting info as well as added experimental data has been refined to not only help communicate our findings, but allow other researchers to use, analyze and interpret our work for future modeling investigations.

Additional notes and remarks:

1) The experiment design and instrumental setup for recording the kinetics of Hartwig-Buchwald aminations has been described by the authors in an earlier publication (Ref. 27).

The earlier publication describing the instrumental setup and Hartwig-Buchwald amination (Ref. 27) examined a much simpler aryl bromide substrate. The main point of the previous publication was to describe and validate the use of this automated sampling platform. The present study leveraged the previously disclosed platform to enable a detailed mechanistic study of a markedly more complex system. The mechanistic distinction between ring-walking and diffusion controlled coupling is impossible to make using the simpler dibromobenzene substrate. The detailed mechanistic insights and application towards synthesizing asymmetric derivatives are entirely novel and, again, would be impossible to conclude from studies with a simpler substrate.

2) Page 3, Fig. 2a: deprotonation should release HBaseX and the resulting intermediate should no longer have X as a ligand.

This error has been fixed and Figure 2A in the manuscript has been updated.

3) Page 9, line 200–201 and Fig. 8, and SI, Suppl. Scheme 2: Here, “diffusion control” implies that the dissociation of Pd(0) and re-association is very rapid (diffusion limit), and that reductive elimination is very rapid [with P(tBu)₃]. Hence, k₁₁ corresponds to the next rate-limiting oxidative addition without ring walk, correct? This may difficult to follow and could be explained in a bit more detail.

We have revised our discussion of our proposed implication around diffusion limited coupling. The model shows that a parallel pathway must exist whereby the catalyst does not escape the

coordination sphere of the Ar substrate. This manifests as a single “ k_{obs} ” and is represented by the theoretical reaction directly providing the tetrasubstituted product directly from the tetra brominated starting material. By no means are we suggesting that this pathway is devoid of the accepted catalytic intermediate required by Pd-catalyzed CN coupling, but simply that a “short-circuit” involving kinetically indistinguishable steps must be at work and operating in parallel to the ring walking behaviour.

4) Page 10, Fig 9: The results only show that, with the PEPPSI catalyst, the second C-N coupling is much faster than the first one (and vice versa with the XantPhos catalyst). Concluding a presence or absence of a ring walk from this change in rates may be overinterpretation. Also: the green label “7” from Fig. 9a got misplaced into Fig 9b.

We have corrected this labeling error and updated Figure 9a in the manuscript. We agree that in theory, one could explain the behavior observed with PEPPSI-IPr as simply arising from an intermediate which displays a significantly increased reactivity compared to the starting material, thus avoiding any buildup under the reaction conditions. To rule out this possibility, we synthesized the monoaminated intermediate 8 and used this as a model system to probe what the rate of coupling would be on such an intermediate with PEPPSI-IPr. Furthermore, we set up such reactions using the same initial concentrations as the parent dibrominated starting material (7). This biases our model system to display faster rates than would ever be observed under standard reaction conditions with 7 where the intermediate is never observed (ie: the [intermediate] remains too low to quantify throughout the reaction). Even under such conditions, the rate of coupling of the 8 is only slightly faster than that observed with the 7. These data strongly suggest that the behavior observed for SpiroOMeTAD can not be explained by vastly different rates of coupling depending on the ligand chosen

5) Page 11, Fig 10 top: Compounds 16a and 16b should not have an identical substitution pattern. Maybe change to 16b: R1/R2/R4 = N(PMP)₂; R3 = Br?

This error has been fixed and Figure 10 in the manuscript has been updated.

6) Page 15, Ref. 28: The citation is missing (journal, volume, page numbers)

This error has been fixed and reference 28 in the manuscript has been updated. The full reference is as follows: Lanni, E. L. and McNeil, A. J. Mechanistic Studies on Ni(dppe)Cl₂-Catalyzed Chain-Growth Polymerizations: Evidence for Rate-Determining Reductive Elimination. *J. Am. Chem. Soc.* 131, 16573-16579 (2009)

7) Supporting Information (general): The SI still contains several (cross)references to Schemes/Figures highlighted in yellow that require updating. This gives the SI an unfinished appearance.

The cross references have been updated, and the remaining yellow highlighting in the SI shows changes made between the originally submitted and resubmitted documents.

8) Supporting Information (general): The concentrations have been determined only indirectly (using solver). I would recommend using calibration curves instead, since the intermediates 4a,b,5 can be prepared and isolated individually (in addition to the substrates and the product).

Alternative methods to calibration curves are well-accepted and commonly used to convert peak area to concentration (J. Org. Chem. 2021, 86, 2, 2012–2016). In the SI we outlined the procedure and assumptions used in our calculation of concentration data and are confident in the validity of this method. To assuage remaining doubts, we have included an overlay of one of the time course reaction profiles with the concentration data solved for using the Solver method and solved for using a traditional calibration curve (Supplementary Figure 14). The two profiles are in extremely good agreement.

We chose to use Solver for the following reasons:

1. Multiple sampling platforms were used to generate time course data for the different substrates and ligands. Each system on each platform would have required a separate calibration curve. The time and resources needed to run so many calibration curves was avoidable given the use of Solver.
2. Similar to point 1, the configuration of each sampling platform underwent minor changes (e.g. replacing a section of tubing) that would have necessitated a new calibration curve. Again to conserve resources and save time, we opted to use Solver.
3. It would have been extremely time consuming to isolate all the intermediates in large enough quantities to run calibration curves. The Solver method is particularly advantageous in systems with many intermediates or byproducts because it circumvents the need for difficult isolations of these materials.

9) Supporting Information (general): The full characterization data (including IR, fragmentation MS, melting point) should be provided for all new compounds.

The compounds have been characterized thoroughly *via* HPLC, ¹HNMR, ¹³CNMR, HMBC, and HRMS data and meet the requirements set by *Nature Communications*. Further we have included X-ray single crystal to authenticate the regiochemistry of the critical disubstituted intermediate, which grounds the central thesis of our study. In addition, we have now included all reaction time course data sets in excel format. This collection represents the single more comprehensive publicly available data set for reaction kinetic analysis associated with any publication at this time.

10) Supporting Information, Fig 3: The experiments with P(*t*Bu)₃ do not seem to be very reproducible, which may render the conclusions drawn from the experiments with this ligand false.

The two standard conditions experiments in Supplementary Figure 3 were run over one year apart using the same solution of P(*t*Bu)₃ in toluene. Unfortunately, we believe the constitution (concentration/degradation) of this solution changes over time resulting in the minor inconsistencies observed with this ligand.

To account for the differences in these reaction profiles we have updated SI Figure 8, which shows overlays of the standard conditions and different excess reactions for the P(*t*Bu)₃ system. This figure now includes both runs of the standard conditions. Even with the margin of error between the standard conditions experiments for P(*t*Bu)₃ the reaction is very clearly slowed when the [1]₀ or [LiHMDS]₀ is reduced. Thus, we feel confident in reporting positive orders in **1** and LiHMDS. The overlays of the reduced [amine]₀ are not as definitive when both standard condition profiles are considered. However, the additional amine dosing experiment shows an undeniable rate increase when [amine]_t is kept low, which is indicative of a negative order. We feel that the different excess experiment in combination with the dosing experiment substantiates our conclusion of a negative order in amine.

Thus, considering the (ir)reproducibility present in the datasets for the ligand did not change or render any of our conclusions false.

11) Supporting Information, Fig 6: As with P(tBu)₃, the experiments do not seem to be 100% reproducible either. The curves for [2] show different start concentrations [2]_{t=0} and a different slope.

The amine in the offline HPLC samples was analytically unstable. This was addressed in the SI: "It should also be noted that **2** was unstable in the offline HPLC samples so the overlays of **2** were not considered in the different excess experiments" (pg S13). The amine overlays were included in Figure S6 to demonstrate this instability and provide clear evidence for why the trends were excluded from the kinetic analyses. We have clarified this point even further in the SI by adding a sentence to the caption of Supplementary Figure 6, "Overlays of [2] are shown to demonstrate the analytical instability of **2**" (pg S9).

12) Supporting Information, Fig 9–11: How can the start concentration of **1** ([1]_{t=0}) be identical if the description states that the concentration was reduced (triangles vs circles)? Are these the correct curves?

We have updated the SI to clarify our data processing of the [1] trends in the reduced [1]₀ experiments. We have added a new figure in the SI (Supplementary Figure 7) and a short description of the data processing, "The concentration versus time profile for **1** in the different excess experiments with reduced [1]₀ have been translated by 20 mM along the y-axis in Supplementary Figure 8, Supplementary Figure 10, Supplementary Figure 12, and Supplementary Figure 13. A clear example of this translation is shown in Supplementary Figure 7. This adjustment is common in visual kinetic analyses and allows for an easier visual comparison of reaction profiles over the same catalyst turnover region (*Chem. Sci.* **10**, 348-353 (2019))" (pg S9). We have also added a clarifying sentence to the relevant figure captions in the SI stating, "See Supplementary Figure 7 for an explanation of why the 'Reduced **1**' trend in the [1] versus time plot starts at 40 mM".

13) Supporting Information, Scheme 1: Since the equilibria are connected, compound **2** should appear on top of the second arrow (as an additive). It is not produced in the first equilibrium. The same applies to the third row.

We have fixed this and updated Supplementary Scheme 1 in the SI.

14) All raw data should be provided in form of tables in the SI or as csv files. The COPASI files used for simulation could be provided as well.

We have included all raw data in the form of Excel spreadsheets (containing the concentration versus time data for each ligand system and aryl halide) and COPASI files (for the mechanistic models).

REVIEWERS' COMMENTS

Reviewer #1 (Remarks to the Author):

I have mixed emotions. The authors addressed some of my critiques from the last round and ignored many others, deeming them inappropriate either b/c of my failings to fully comprehend or appreciate their work, or because it would broaden the scope of the paper too much to include it.

Nevertheless, the ones that were addressed (from me and others) have made the paper better and they have fixed many errors in their original draft. I still have unease about the sweeping conclusions from the data provided, and it is true that some of this skepticism is focused on the kinetics method used.

My misgivings are largely focused on the interpretation of rates from the visual analysis of concentration versus time profiles. As an example, let's look at the case of PtBu₃. The authors conclude a "negative" order in 2 but the difference in rate for "reduced 2" vs "std conditions" in the plot for formation of [6] (product – bottom plot in blue) is as big as the error in the two different runs for "std conditions" – see blue curve below. I can't believe you can conclude it's faster with reduced 2 when your error is this big. In another example, for XantPhos, the data seems to indicate inverse or negative order in [1] bc the reduced [1] curves look faster (most clear in purple and blue plots below), yet the authors conclude zeroth order from this same data.

I'll admit I'm not a fan of the COPASI modeling as multiple elementary steps are combined into one "step" represented with a single rate constant. Here I'll focus on the RuPhos model. (I'm not sure what "int-1" is, actually, b/c the paper refers to it as ArBr_cat and I'm not sure if that's just binding (as labeled here) or after oxidative addition.) Either way, k₂ represents (maybe) oxidation addition, coordination, deprotonation, and reductive elimination x 2 and k₄ represents again these four steps x 2. So... what are these rate constants really? For the slow step only (proposed reductive elimination) and that's where you derive your units?

One minor comment: Figure 3a is quite confusing. In path 1, are you suggesting that its ring-walking and diffusion control operating as parallel, competing pathways b/c on my first and second glance, it looks like you are proposing two types of ring-walking where one has diffusion control and one does not, which doesn't make sense to me nor is it consistent with your scheme in Figure 2. Also – why are there four pathways here in Figure 3 and only 3 pathways in Figure 2. Maybe this is where the confusion starts.

Finally, there is a lot of commentary on "resting states" in the intro, paper, and conclusion but again, the authors did not characterize ANY resting states in this work. They hypothesize what they might be based on their limited data. I think they should remove or modify these sentences. For example, in the conclusion "our results suggest that the resting state of the catalyst does little to promote or inhibit ring walking" – also, as an aside, there is no reason to believe why it would.

Reviewer #2 (Remarks to the Author):

Revisions made by the authors have significantly improved the robustness and quality of this manuscript. The authors' responses to all of the comments raised by this reviewer are satisfactory and well appreciated for their level of detail. All of my concerns have been thoroughly addressed in the new manuscript / SI. I thank the authors for their effort to improve the clarity and depth of their report, and for the additional experimental work undertaken in doing so.

It is my recommendation to accept this manuscript for publication in this current revised condition.

One very minor typo: in paragraph 'COPASI modelling', there are two references to 'figure 8b', one of which

should say 'figure 8a'.

Reviewer #3 (Remarks to the Author):

In the revised version, several issues raised by the reviewers have been addressed. The introduction is easier to follow and I appreciate that the modeling part has been revised as well. It now includes details on the models that were applied and some of the valuable and insightful comments by reviewer 2 have been considered in the revision of the manuscript and the supporting information. This has significantly improved the quality of the manuscript. For the sake of conciseness, I will not comment in detail on the discussion of reviewer 2's remarks and the answers by the authors.

I agree with the authors that most of the criticism raised by reviewer 1 is not justified. The results from this study are in agreement with the proposed scenarios (ring walking / not ring walking) for the different ligands. Since the current study is solely based on kinetics involving non-metalated intermediates, further support could have been added through additional information about which Pd intermediates are detectable, yes. But in general, the data and modeling results are in agreement with the conclusions. Other scenarios cannot be 100% ruled out, in particular in the cases where a fairly complex model is required to accurately model the data (with enough equations, one can model any system). On the other hand, the Pd ring walking on such systems has been studied and confirmed before, and I cannot think of such a different scenario that is in agreement with the observations. Regarding the determination of orders, it can be done the way the authors have done it in order to determine zeroth order or pos./neg. order dependencies. The information gathered is sufficient for this study. In the case of non-zeroth orders, one could have determined the orders more precisely via RPKA (reaction progress kinetics analysis, rate vs [P] plot) or VKA (visual kinetic analysis, see the mentioned Chem. Sci. Review by Nielsen and Bures) to improve the quality of the study. But the presented results are sufficient to get the information needed. Nevertheless, JACS 2018, 140, 7846 could be cited in the context of preceding studies on such ring walking reactions.

Regarding the response to my own comments (reviewer 3), I agree that the earlier study by these authors did not allow the differentiation between diffusion controlled and ring-walking coupling. The current substrate design is cleverly chosen to probe this (reviewer 2 also praised this choice of substrate as well) and the results are probably important for the area of polymerization chemistry. But I am not convinced that the current study is of ground breaking relevance to catalysis and other fields in general. The manuscript itself seems more like a follow-up study of the authors earlier publication on ring walking, now applying the technique to a more complex substrate from the same class of compounds that were studied earlier. I agree that this study establishes a new way of regiocontrol for Hartwig-Buchwald couplings (and probably also other palladium catalyzed couplings) that cannot be achieved otherwise. But it is overall an application and elaboration of earlier findings paired with a detailed kinetic analysis. Considering that this is not the first study by the authors on such a ring walking reaction using this sampling technique, the work could have revealed more conclusions on how the ligand structure affects the ring walking behavior (I agree with reviewer 1 in this point) in order to be published in a top tier journal such as Nature Communications.

Overall, this work has gained quality and it is publishable. The conclusions are in agreement with the presented data. But, I am not sure whether the insight from this work will be of relevance to a broad, cross-discipline readership, because it applies only to a particular group of substrates that is mainly used in polymer chemistry. I therefore recommend publication in a more specialized journal.

REVIEWERS' COMMENTS

Reviewer #1 (Remarks to the Author):

I have mixed emotions. The authors addressed some of my critiques from the last round and ignored many others, deeming them inappropriate either b/c of my failings to fully comprehend or appreciate their work, or because it would broaden the scope of the paper too much to include it.

Nevertheless, the ones that were addressed (from me and others) have made the paper better and they have fixed many errors in their original draft. I still have unease about the sweeping conclusions from the data provided, and it is true that some of this skepticism is focused on the kinetics method used.

My misgivings are largely focused on the interpretation of rates from the visual analysis of concentration versus time profiles. As an example, let's look at the case of PtBu₃. The authors conclude a "negative" order in 2 but the difference in rate for "reduced 2" vs "std conditions" in the plot for formation of [6] (product – bottom plot in blue) is as big as the error in the two different runs for "std conditions" – see blue curve below. I can't believe you can conclude it's faster with reduced 2 when your error is this big. In another example, for XantPhos, the data seems to indicate inverse or negative order in [1] bc the reduced [1] curves look faster (most clear in purple and blue plots below), yet the authors conclude zeroth order from this same data.

We agree with the reviewer that the error observed in PtBu₃ is more significant than that observed in the case of the other 3 ligands studied. However, to further support the conclusion of a negative order in 2, a dosing experiment was conducted where a significant reduction in reaction time was observed further corroborating this observation (see supplementary figure 9).

The behavior observed in the XantPhos data is more consistent with catalyst decay/inhibition than that of a change in order. This can be gleaned by the fact that the rate of formation of each intermediate overlays very well up until the formation of the final intermediate and product. At which point deviation can be observed with respect to the reduced ArBr data. This can be rationalized by the fact that, at this point, the reduced ArBr reaction has undergone significantly less turnovers, and thus may suffer from less catalyst decay/inhibition relative to standard reaction conditions. If there was in fact a negative order in 1 as the reviewer suggests, we would see this manifest from the beginning observing rates of decay of 1, and rates of formation of the various intermediates significantly faster than that of standard conditions.

I'll admit I'm not a fan of the COPASI modeling as multiple elementary steps are combined into one "step" represented with a single rate constant. Here I'll focus on the RuPhos model. (I'm not sure what "int-1" is, actually, b/c the paper refers to it as ArBr_cat and I'm not sure if that's just binding (as labeled here) or after oxidative addition.) Either way, k₂ represents (maybe) oxidation addition, coordination, deprotonation, and reductive elimination x 2 and k₄ represents again these four steps x 2. So... what are these rate constants really? For the slow step only (proposed reductive elimination) and that's where you derive your units?

We agree with this reviewer that conclusions derived from COPASI modeling must be derived with a high level of care. As stated in the paper, best practice when using COPASI is to begin with the simplest possible model while adding in complexity one step at a time until the model matches. Using this procedure, we are able to strongly support the presence of competitive diffusion controlled coupling. This is the main point of our copassi work and as such is highlighted directly in the body of the paper. We agree that reporting of quantitative k values using such a model is not appropriate in such a system. As a result, our discussion of the relevance of the observed k values remains qualitative, as a manner to support our identification of the minimal model (ie: given that our k values share similar magnitudes, this suggests our minimal model is ideal and doesn't suffer from overfitting). However, this reviewer's comment suggests we have not made this clear enough and therefore have added a sentence addressing this issue in the supplementary information.

One minor comment: Figure 3a is quite confusing. In path 1, are you suggesting that its ring-walking and diffusion control operating as parallel, competing pathways b/c on my first and second glance, it looks like you are proposing two types of ring-walking where one has diffusion control and one does not, which doesn't make sense to me nor is it consistent with your scheme in Figure 2. Also – why are there four pathways here in Figure 3 and only 3 pathways in Figure 2. Maybe this is where the confusion starts.

Figure 2 is meant to set up the discussion of the difficulties in identifying authentic ring walking vs mechanistic regimes which would manifest identical kinetics. Three pathways which have been proposed in the literature are highlighted: a) authentic ring walking, b) increased reactivity of the intermediate, and c) diffusion controlled coupling. We stress that path b can be successfully identified using time course data however, differentiation between path a and c remains, to our knowledge, impossible as they would manifest identical kinetics. With that background, figure 3 is made to emphasize why the choice of the tetrabrominated spirocyclic model system is key in enabling the identification of which path (a or c) is operative under the reaction conditions. Here we delineate what each time course would look like if none were operative, if only diffusion controlled coupling were operative, if only ring walking was operative, and if both were operative. This results in the 4 potential scenarios shown in figure 3a.

Finally, there is a lot of commentary on “resting states” in the intro, paper, and conclusion but again, the authors did not characterize ANY resting states in this work. They hypothesize what they might be based on their limited data. I think they should remove or modify these sentences. For example, in the conclusion “our results suggest that the resting state of the catalyst does little to promote or inhibit ring walking” – also, as an aside, there is no reason to believe why it would.

We strongly disagree with these comments from the reviewer. The use of kinetic analysis such as that presented to infer resting state is well accepted and is observed time and time again in various studies using initial rates, RPKA, and more recently VTNA. Moreover, as highlighted in the prior rebuttal letter, our results also match those of Buchwald et al. when they conducted their kinetic study of diarylamine reductive elimination. This provides further corroboration that our analysis of resting states for this particular study is well founded. Finally, although not explicitly stated in the body of the paper, the reason why one may expect a significant impact from catalyst resting state on the manifestation of ring

walking is the following: If oxidative addition is rate determining and the resting state of the catalyst is Pd(0), it's reasonable to assume that the barrier for the Pd(0) to come unbound from the pi system or undergo intermolecular transfer to another pi system would be low. In such cases ring walking would not be observed. It's also worth highlighting that the kinetic studies thus far on ring walking polymerization have observed resting states in the 2+ oxidation state.

Reviewer #2 (Remarks to the Author):

Revisions made by the authors have significantly improved the robustness and quality of this manuscript. The authors' responses to all of the comments raised by this reviewer are satisfactory and well appreciated for their level of detail. All of my concerns have been thoroughly addressed in the new manuscript / SI. I thank the authors for their effort to improve the clarity and depth of their report, and for the additional experimental work undertaken in doing so.

It is my recommendation to accept this manuscript for publication in this current revised condition.

One very minor typo: in paragraph 'COPASI modelling', there are two references to 'figure 8b', one of which should say 'figure 8a'.

We thank this reviewer for their comments and have since addressed this typo in the text of the manuscript.

Reviewer #3 (Remarks to the Author):

In the revised version, several issues raised by the reviewers have been addressed. The introduction is easier to follow and I appreciate that the modeling part has been revised as well. It now includes details on the models that were applied and some of the valuable and insightful comments by reviewer 2 have been considered in the revision of the manuscript and the supporting information. This has significantly improved the quality of the manuscript. For the sake of conciseness, I will not comment in detail on the discussion of reviewer 2's remarks and the answers by the authors.

I agree with the authors that most of the criticism raised by reviewer 1 is not justified. The results from this study are in agreement with the proposed scenarios (ring walking / not ring walking) for the different ligands. Since the current study is solely based on kinetics involving non-metalated intermediates, further support could have been added through additional information about which Pd intermediates are detectable, yes. But in general, the data and modeling results are in agreement with the conclusions. Other scenarios cannot be 100% ruled out, in particular in the cases where a fairly complex model is required to accurately model the data (with enough equations, one can model any system). On the other hand, the Pd ring walking on such systems has been studied and confirmed before, and I cannot think of such a different scenario that is in agreement with the observations. Regarding the determination of orders, it can be done the way the authors have done it in order to determine zeroth order or pos./neg. order dependencies. The information gathered is sufficient for this study. In the case of non-zeroth orders, one could have determined the orders more precisely via RPKA (reaction progress kinetics analysis, rate vs [P] plot) or VKA (visual kinetic analysis, see the mentioned

Chem. Sci. Review by Nielsen and Bures) to improve the quality of the study. But the presented results are sufficient to get the information needed. Nevertheless, JACS 2018, 140, 7846 could be cited in the context of preceding studies on such ring walking reactions.

We agree with this reviewer's comments and have added the reference suggested in the body of the paper.

Regarding the response to my own comments (reviewer 3), I agree that the earlier study by these authors did not allow the differentiation between diffusion controlled and ring-walking coupling. The current substrate design is cleverly chosen to probe this (reviewer 2 also praised this choice of substrate as well) and the results are probably important for the area of polymerization chemistry. But I am not convinced that the current study is of ground breaking relevance to catalysis and other fields in general. The manuscript itself seems more like a follow-up study of the authors earlier publication on ring walking, now applying the technique to a more complex substrate from the same class of compounds that were studied earlier. I agree that this study establishes a new way of regiocontrol for Hartwig-Buchwald couplings (and probably also other palladium catalyzed couplings) that cannot be achieved otherwise. But it is overall an application and elaboration of earlier findings paired with a detailed kinetic analysis. Considering that this is not the first study by the authors on such a ring walking reaction using this sampling technique, the work could have revealed more conclusions on how the ligand structure affects the ring walking behavior (I agree with reviewer 1 in this point) in order to be published in a top tier journal such as Nature Communications.

Overall, this work has gained quality and it is publishable. The conclusions are in agreement with the presented data. But, I am not sure whether the insight from this work will be of relevance to a broad, cross-discipline readership, because it applies only to a particular group of substrates that is mainly used in polymer chemistry. I therefore recommend publication in a more specialized journal.

We appreciate this reviewer's comments however respectfully disagree that this work is not of broad relevance. First and foremost, our prior work on Buchwald Hartwig couplings and ring walking was necessary to support the reliability and reproducibility of our automated sampling platform. The conclusions drawn with respect to ring walking were similar to those already established in the literature. In contrast, the current work serves to provide the first direct kinetic evidence of ring walking under standard reaction conditions. The ability to differentiate between ring walking and diffusion controlled coupling further increases the impact of this work as this diagnostic tool could be leveraged to develop and optimize catalysts towards either mechanistic regime.

As highlighted by the reviewer, this work clearly impacts polymerization chemistry, as the ability to detect and understand ring walking is of the utmost importance. However, this work also increases the breadth of chemical space easily accessed for the development of small molecule materials as highlighted at the end of the manuscript. We were able to harness unprecedented selectivity to provide direct access to a valuable building block in a single step. This has broad implications on synthesizing and screening compound libraries for material optimization which often make use of polyhalogenated starting materials. Moreover, selectivity in polyhalogenated settings for the synthesis of bioactive compounds has also attracted much attention as of late (Chem. Rev. 2021,

<https://doi.org/10.1021/acs.chemrev.1c00513>). We expect this work highlighting the nuances between ring walking and diffusion control to help guide the identification and development of highly selective catalysts.